

# Parameterized minimum eddy diffusivity in WRF-Chem(v3.9.1.1) for improving PM$_{2.5}$ simulation in the stable boundary layer over eastern China

Wen Lu[1,2,3], Bin Zhu[1,2,3*], Shuqi Yan[4], Jie Li[5,6], Zifa Wang[5,6]

[1]Collaborative Innovation Center on Forecast and Evaluation of Meteorological Disasters, Nanjing University of Information Science & Technology, Nanjing 210044, China
[2]Key Laboratory for Aerosol-Cloud-Precipitation of China Meteorological Administration, Nanjing University of Information Science & Technology, Nanjing 210044, China
[3]Key Laboratory of Meteorological Disaster, Ministry of Education (KLME), Nanjing University of Information Science & Technology, Nanjing 210044, China
[4]Key Laboratory of Transportation Meteorology of China Meteorological Administration, Nanjing Joint Institute for Atmospheric Sciences, Nanjing 210041, China
[5]LAPC, Institute of Atmospheric Physics, Chinese Academy of Sciences, Beijing, China
[6]University of Chinese Academy of Sciences, Beijing, China

*Correspondence to*: Bin Zhu (binzhu@nuist.edu.cn)

**Abstract.** Weak turbulence often occurs during heavy pollution events in eastern China. However, existing mesoscale models cannot accurately simulate turbulent diffusion under weakened turbulence, particularly under the nocturnal stable boundary layer (SBL), often leading to significant turbulent diffusivity underestimation and surface aerosol simulation overestimation. In this study, based on the Weather Research and Forecasting model coupled with the Chemistry model (WRF-Chem 3.9.1), a new parameterization of minimum turbulent diffusivity (Kzmin) is tested and applied in PM$_{2.5}$ simulations in eastern China under SBL conditions. Sensitivity experiments show that there are different value ranges of available Kzmin over the northern (0.8 to 1.3 m$^2 \cdot$s$^{-1}$) and southern (1.0 to 1.5 m$^2 \cdot$s$^{-1}$) regions of East China. The geographically related Kzmin could be parameterized by means of two factors: sensible heat flux (H) and latent heat flux (LE), which also exhibited a regional difference related to the climate and underlying surface. The revised Kzmin scheme obviously enhanced the turbulent diffusion (north: 0.88 m$^2 \cdot$s$^{-1}$, south: 1.17 m$^2 \cdot$s$^{-1}$ on average) under the SBL, simultaneously improving the PM$_{2.5}$ simulations, with the PM$_{2.5}$ relative bias decreasing from 44.0% to 14.16% on the surface. The improvement in the mean bias of the surface simulation was more noticeable in the north (57.99 to 3.43 ug$\cdot$m$^{-3}$) than in the south (37.77 to 19.8 ug$\cdot$m$^{-3}$). It also increased the PM$_{2.5}$ concentration in the upper SBL. Furthermore, we discussed the physical relationship between Kzmin and two factors. Kzmin was inversely correlated with sensible heat flux (negative) and latent heat flux (positive) in the SBL. Process analysis showed that vertical mixing is the key process to improve PM$_{2.5}$ simulations on the surface in the revised scheme. The increase in the PM$_{2.5}$ concentration in the upper SBL was attributed to vertical mixing, advection, and aerosol chemistry. This study highlights the importance of improving turbulent diffusion in





current mesoscale models under the SBL and has great significance for aerosol simulation research under heavy air pollution

events.

## 1 Introduction

The aerosol particles with aerodynamic diameters less than 2.5 µm ($PM_{2.5}$), which is still the principal air pollutant over eastern China in wintertime, has received widespread attention in recent decades (Cai et al., 2017; Li et al., 2017; Hou et al., 2019; Liu et al., 2021). Numerical models are an important tool to study evolutionary mechanisms. However, there are

serious shortcomings in $PM_{2.5}$ simulations during rapid growth and severe air pollution events (Wang et al., 2018), especially under the stable boundary layer (SBL). An accurate emissions inventory is an indispensable factor in obtaining accurate $PM_{2.5}$ simulations. Meteorological conditions, particularly turbulent diffusion, play a critical role in the evolution of pollutants in the boundary layer (BL) when emissions are constant on a monthly scale (Kurata et al., 2004; Sofiev et al., 2013; Jia et al., 2021b; Liu et al., 2022).

Studies of the SBL remain insufficient; the SBL is often accompanied by intermittent turbulence and decoupling of the surface and free troposphere (Louis 1979; Grachev et al. 2005). At night, buoyancy is typically weak, which could lead to the calculation of a zero-turbulence diffusivity value in the model (Li et al., 2018). The diffusion of air pollutants is governed by the minimum turbulent diffusivity (Kzmin), which is determined from the planetary boundary layer (PBL) scheme (Li et al., 2018). Huang et al. (2010) improved the turbulent fluxes in the SBL by redefining the closure constants and modifying

the sensible heat flux prognostic equation. Jia et al. (2021a) developed a novel formula for particle diffusion based on mixing-length theory. It has improved the simulation of aerosols in the SBL over eastern China. In addition to updating the turbulent diffusion formula, some modifications were focused on altering the value of Kzmin. Li et al. (2018) and Ding et al. (2021) parameterized Kzmin by land-use category. The value of Kzmin depends on the urbanization rate on land, while it is 0 on water. They investigated the model performance between fixed values and parameterized values. In comparison to the

fixed value, ozone biases that are simulated by the parameterized value are 31% lower, and the simulation of temperature has been improved. Du et al. (2020) and Wang et al. (2021) evaluated the simulation performance of aerosols by using different fixed Kzmin values in WRF-Chem and found that the higher the values of Kzmin were, the closer to the observations they were, particularly under the nocturnal SBL. Because the values of Kzmin were different based on regions and events in previous research, adopting a fixed value of Kzmin for a large region (e.g., east of China) and long-term (e.g., the whole

month) study might not be an option. Therefore, a flexible Kzmin value is required for BL meteorology and aerosol simulations.

This study aims to improve the model performance of aerosol and BL temperature simulations in the nocturnal SBL by implementing a novel parameterized scheme of Kzmin. Additionally, we attempt to discuss the physical meanings of Kzmin. Furthermore, we attempt to determine the key process in the simulation improvements by process analysis technology. Our

research results would be useful for the improvement of $PM_{2.5}$ and BL-Met simulations in the nocturnal SBL over eastern





China and will be of great significance for aerosol research across this region. The rest of this paper is organized as follows. The description of the observation data, model configuration and numerical experiments will be presented in section 2. The results and discussion will be presented in section 3, and the summary can be found in section 4.

## 2 Data and Methodology

### 2.1 Observation data


Three sets of data were used to evaluate model performance. The first set of data is the hourly ground-based observations of PM$_{2.5}$ mass concentrations in 89 cities obtained from the China National Environmental Monitoring Center and published online (http://106.37.208.233:20035). The second set of data is the 3 h-hourly meteorological factors at 99 ground observation stations in eastern China. The meteorological factors contain 10 m wind speed, 10 m wind direction, and

2 m temperature. The third set of data is the vertical observations of PM$_{2.5}$ and meteorological factors from field experiments by our group in Nanjing. The field experiment was carried out between 27 December 2016 and 31 December 2016 to obtain the 3 h-resolution vertical distribution data of PM$_{2.5}$.

### 2.2 Model configuration

The WRF-Chem model (V3.9.1.1), a fully coupled online 3-D Eulerian meteorological and chemical transport model,

was used in this study. It has been widely applied in air quality research (Grell et al., 2005; Lu et al., 2023). The parent domain (D01) has 99×99 grids with a resolution of 27 km, covering most parts of China and the surrounding regions and ocean. The nested domain (D02) has 129×150 grids and a 9 km resolution covering most of East China (Fig. 1). Thirty-eight layers were set up from the surface up to the 50 hPa level, of which 13 layers were located below the lowest 2 km to explore the precise BL structure. The PBL scheme is YSU, which is a nonlocal closure BL scheme(Hong et al. 2006). The Carbon-

Bond Mechanism version Z (CBMZ) and the Model for Simulating Aerosol Interactions (MOSAIC) with aqueous chemistry were chosen for gas-phase and aerosol chemical processes (Zaveri et al., 2008; Zaveri and Peters et al., 1999). Other parameterization schemes for physical processes included the Morrison 2–moment Scheme (Morrison et al., 2009), the rapid radiative transfer model for GCM (RRTMG) short- and longwave radiation (Iacono et al., 2008), and the Noah land surface model (Chen et al., 2001).


The model was driven by the National Centers for Environmental Prediction final (FNL) data file product (1°×1°, https://rda.ucar.edu/datasets/ds083.2/index.html, last accessed: September 18, 2022). The chemical initial and boundary conditions were provided by the Community Atmosphere Model outputs with Chemistry (CAM-chem; Emmons et al., 2020). The anthropogenic emissions are derived from the Multiresolution Emission Inventory for China (MEIC; emission index year is 2016; Li et al, 2017a; Zheng et al, 2018) database and MIX (emission index year is 2010; Li et al, 2017b). The

model's biogenic emissions were generated by the Model of Emission of Gas and Aerosols from Nature (MEGAN; Guenther



et al., 2006). Based on the above model configurations and observational data, we designed the new Kzmin scheme in section 3.1 and compared and evaluated the results of the new and old schemes.

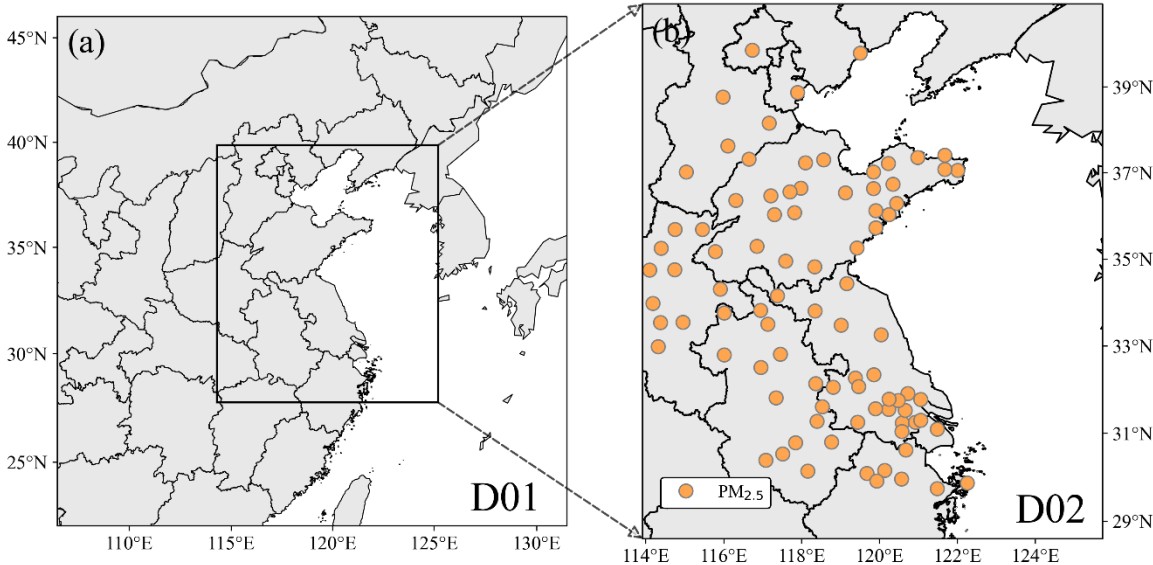

**Figure 1.** The model domain of D01(a) and D02(b). Air quality monitoring sites (i.e., 89 stations) and their locations are shown in (b), used for model validation.

## 2.3 Numerical experiments

The turbulent mixing of pollutants in WRF-Chem model is a process related to the turbulence diffusion coefficient and the concentration difference between vertical adjacent (above or below) grids. The turbulent mixing process of pollutants is considered to be similar to that of heat, which supposes the turbulent diffusion of particles and heat is identical (Jia et al., 2021b). In this study, we adopted the YSU scheme as the PBL scheme due to it is widely used in eastern China (Du et al., 2020, Gao et al., 2022, Yan et al., 2023). Compared to the previous generation version MRF scheme, YSU scheme increased vertical mixing in the buoyancy driven and decreased it in the mechanic driven (Hong et al., 2006). In YSU sheme, the momentum mixing coefficient $K_m$ in the mixed layer is formulated following Hong et al. (2006):

$$K_m = kw_s z(1 - z/h)^p \tag{1}$$

where k is the von Karman constant (=0.4), $w_s$ is the mixed layer velocity scale, z is the height from the surface, h is the height of the PBL, and p is the profile shape exponent taken to be 2. According to the Prandtl number as in Noh et al., 2003:

$$P_r = 1 + (P_{r0} - 1)\exp\left[\frac{-3(z - \varepsilon h)^2}{h^2}\right] \tag{2}$$

the heat mixing coefficient $K_h$ can be calculated following:

$$K_h = K_m/P_r + Kzmin \tag{3}$$



where Kzmin is the minimum eddy diffusivity ($=0.01$ m$^2 \cdot$s$^{-1}$).

However, many studies have suggested that mesoscale models underestimate the mixing in SBL simulation and lead to the overestimation of PM$_{2.5}$ simulation (Teixeira et al. 2008, Du et al., 2020, Jia et al., 2021a). Therefore, two sets of experiments are designed in our research. In the control experiment (EXP_BASE), the default value of Kzmin in the YSU scheme is set to 0.01 m$^2 \cdot$s$^{-1}$. Through sensitivity experiments on Kzmin (shown in Table S1, the available values are marked

in red), we found that the available Kzmin values for winter aerosol simulations are different in the north of eastern China (NCP: 0.8 to 1.3 m$^2 \cdot$s$^{-1}$) and in the Yangtze River Delta (YRD:1.0 to 1.5 m$^2 \cdot$s$^{-1}$). The sensible heat flux (H), also known as sensible heat transfer, refers to the turbulent heat exchange that occurs between the atmosphere and the underlying surface due to temperature differences. The latent heat flux (LE) is usually used to characterize the heat released or absorbed by a substance during the phase change of water. Some studies have suggested that the spatial distributions of H and LE are

related to climate zones and different underlying surfaces (Dan et al., 2011). The evaporative fraction (EF) was used to represent the relative contributions of the turbulent energy fluxes to the surface energy budget (Shuttleworth et al., 1989), which is defined as the ratio of the LE to the sum of the H and LE:

$$EF = LE/(H + LE) \tag{4}$$

We assume that the value of EF can be used to characterize Kzmin in different regions. As such, we parameterized the new

value of Kzmin in the PBL scheme by that in EXP_NEW, and the expression can be found in formula 1.

$$under\ stable\quad : Kzmin = EF + 1.0 \tag{5}$$

$$under\ unstable : Kzmin = 0.01 \tag{6}$$

When the grid in the PBL was under stable conditions (Ri > 0), the Kzmin value was set to the value calculated by formula 1. While the grid in the PBL was under unstable conditions (Ri<0), the Kzmin value was set to the default value

(0.01). To avoid outlier calculation results, we set the Kzmin value variations from 0.01 to 2.0 (93% grid values fall within this interval). By comparing EXP_BASE with EXP_NEW, we can explore the impact of Kzmin on the PM$_{2.5}$ simulation. We will also discuss the physical relationships of Kzmin with EF in section 3.2.

## 3 Results and discussion

### 3.1 Evaluation of the new scheme

In this study, model performance metrics (MB, mean bias; IOA, index of agreement; RMSE, root mean square error; R: correlation coefficient, NMB: normalized mean bias, NME: normalized mean error) were used to validate meteorological factors and air pollution (T$_{2m}$: temperature at 2 m above the surface; WS$_{10m}$, wind speed at 10 m above the surface; WD$_{10m}$, wind direction at 10 m above the surface and PM$_{2.5}$ on the surface).





### 3.1.1 Simulation of meteorological factors and PM$_{2.5}$ on the surface

The mean model performance of the meteorological factors and PM$_{2.5}$ in EXP_BASE is shown in Table 1. T$_{2m}$ showed high values of the mean IOA and R, which indicated that the simulation agreed well with the observations. Approximately 82% of stations underestimated T$_{2m}$, which caused the MB mean value to be negative (-0.86 °C). The simulation of WS$_{10m}$ was slightly higher than the observation, while the IOAs and RMSEs met the criteria. The mean IOA of WD$_{10m}$ reached 0.87, which also suggests good agreement between the wind simulation and observation. In general, the model satisfactorily

captured the variation in meteorological factors.

**Table 1.** Mean model performance metrics for meteorological factors and PM$_{2.5}$ (daytime and nighttime) in EXP_BASE. The values that unsatisfied the EPA suggested are highlighted in bold.

| Variable | MB | IOA | RMSE | R |
|---|---|---|---|---|
| T$_2$ | -1.15 [-0.5,0.5]) | **0.82** (≥0.8) | 2.63 | 0.81 |
| WS$_{10m}$ | 0.60 ([-0.5,0.5]) | **0.69**(≥0.6) | **1.46**(≤2) | 0.6 |
| WD$_{10m}$ | **2.85**([-10,10]) | 0.87 | 64.12 | 0.87 |
| PM$_{2.5}$_day | 19.57 | 0.77 | 65.35 | 0.7 |
| PM$_{2.5}$_night | 48.23 | 0.72 | 82.68 | 0.66 |

The simulation of PM$_{2.5}$ was overestimated both in the daytime (8:00 to 16:00) and nighttime (19:00 to 5:00 the next day), and the overestimation was more obvious at nighttime. The mean MBs at nighttime are more than twice as high as those during the daytime. The mean IOAs, RMSEs, and Rs during the daytime are also better than those at night. In short, the daytime simulations are significantly better than the nighttime simulations, which may be related to the good performance of the boundary layer mechanism in the model under the convective boundary layer. The simulation under

nocturnal SBL needs to be improved. Therefore, we will focus on the improvement during the nocturnal SBL in our research.

The distribution of the simulation (shaded) and observation (scatter) from each station is shown in Figure 2a (EXP_BASE). Obviously, the model overestimated the simulation of PM$_{2.5}$ both in the north and south, which is similar to previous research in this region by WRF-Chem (Du et al., 2020; Jia et al., 2021a). A total of 96.9% of stations overestimate the simulation concentration with a mean relative bias equal to 44.0%. The deviation is larger in the inland regions than in

the coastal regions. The relative simulation bias of PM$_{2.5}$ is slightly different in the north (NCP: Beijing, Tianjin, Hebei, Henan and Shandong) and south (YRD: Anhui, Jiangsu, Shanghai and Zhejiang), and the north is more overestimated. This may be related to geographical conditions, climate and emission differences, and the degree of pollution (Liu et al., 2022; Wang et al., 2018). In comparison to the YRD, the degree of overestimation is more apparent in the NCP. The mean value of MB was 57.99 µg·m$^{-3}$ in the NCP and 37.77 µg·m$^{-3}$ in the YRD (Figure 2b, c). The mean IOA and R in the NCP are higher

than those in the YRD. Thus, the simulated variation trend is closer to the observation in the NCP than in the YRD.



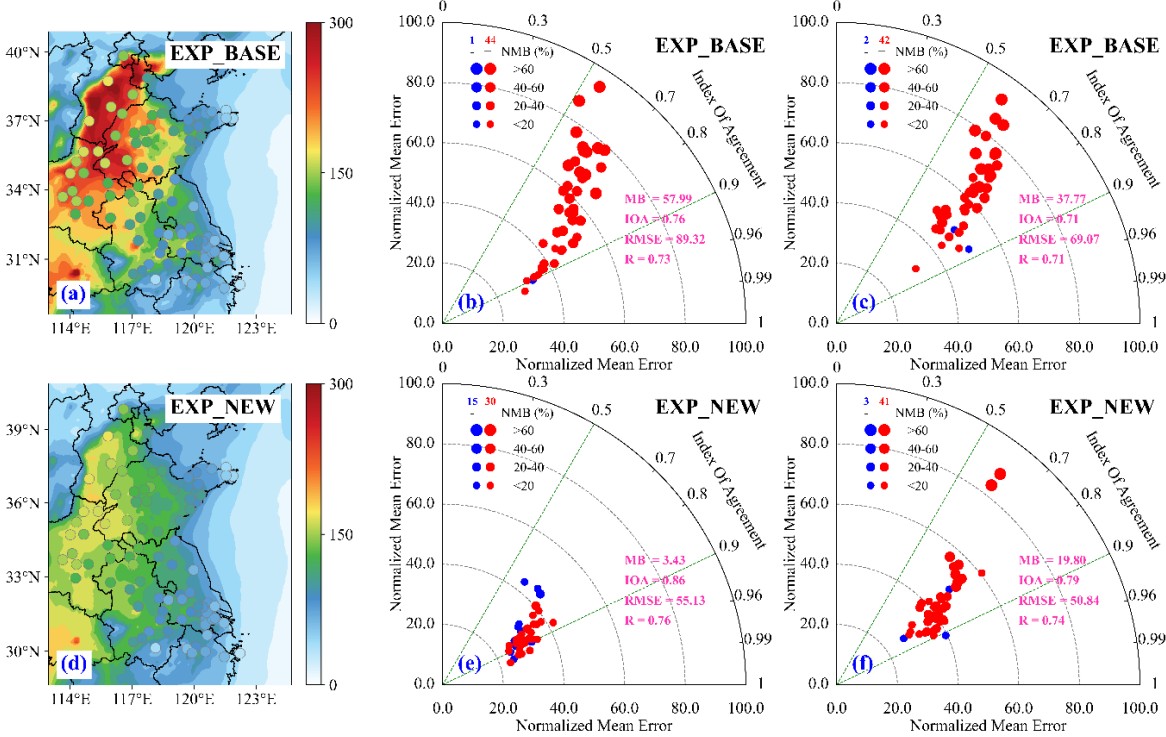

**Figure 2.** The distribution of PM$_{2.5}$ concentration (unit: μg·m$^{-3}$) in EXP_BASE (a) and EXP_NEW(d). The shaded and scatter are represented for simulation and observation, respectively. Taylor diagram for displaying model performance metrics of PM$_{2.5}$ simulation in EXP_BASE (b: NCP; c: YRD) and EXP_NEW (e: NCP; f: YRD). The radial distance from the origin, the azimuthal position and the size of dots represent the values NME, IOA, and NMB of each station, respectively. The mean model performance was shown by the pink font.

A new PBL scheme was introduced in EXP_NEW and solved the overestimation in eastern China (Figure 2d). The performance of the meteorological factor simulation is shown in Figure S1. The simulation of temperature has a better enhancement, which is consistent with the results of the study by Ding et al. (2021). For wind, there is not much difference between EXP_NEW and EXP_BASE. For PM$_{2.5}$, the mean relative bias of the PM$_{2.5}$ simulation decreased from 44.0 % to 14.1%. The mean value of MB decreased to 3.43 μg·m$^{-3}$ in the NCP, and most NME values clustered between 20 and 40 (Figure 2e). The mean value of R (IOA) increased from 0.73 (0.76) in EXP_BASE to 0.76 (0.86). For the YRD, the improvement effect of the PM$_{2.5}$ simulation is not as significant as that in the NCP (Figure 2f). The value of NME clustered between 20 and 60 with a larger mean MB (19.8 ug·m$^{-3}$). The mean value of R (IOA) increased from 0.71 (0.71) in EXP_BASE to 0.74 (0.79). The mean IOA and R values in the YRD have increased significantly, indicating that the introduction of Kzmin may make the simulation more consistent with the observed trend (Figure 2f). Although there is no significant improvement in the mean MB in the YRD, the simulated trend is more similar to the observation. Therefore, we believe that the simulation in the YRD has also been improved.




### 3.1.2 The simulation of PM$_{2.5}$ in the vertical direction

Aerosols have a significant impact on the radiation and structure of the BL in the vertical direction (Yan et al., 2023). We further evaluate the vertical simulation of PM$_{2.5}$ during nighttime over eastern China. The difference shows that EXP_NEW decreased the concentration of PM$_{2.5}$ on the surface, especially in the NCP region (Figure 3b). The vertical section from the NCP to the YRD (Figure 3a) showed that EXP_NEW decreased the PM$_{2.5}$ concentration on the surface and increased it in the upper BL (approximately 0.5-1.2 km). The results of UAV vertical sounding show that EXP_NEW makes

the vertical simulation more reasonable (Figure 4). The simulation in EXP_BASE overestimated the PM$_{2.5}$ concentration on the surface and underestimated it in the upper BL.

In general, EXP_NEW can enhance the simulation ability in the meteorological factor and PM$_{2.5}$ in the nocturnal SBL better than EXP_BASE. In the next section, we attempt to explain the physical effects of the newly designed Kzmin.

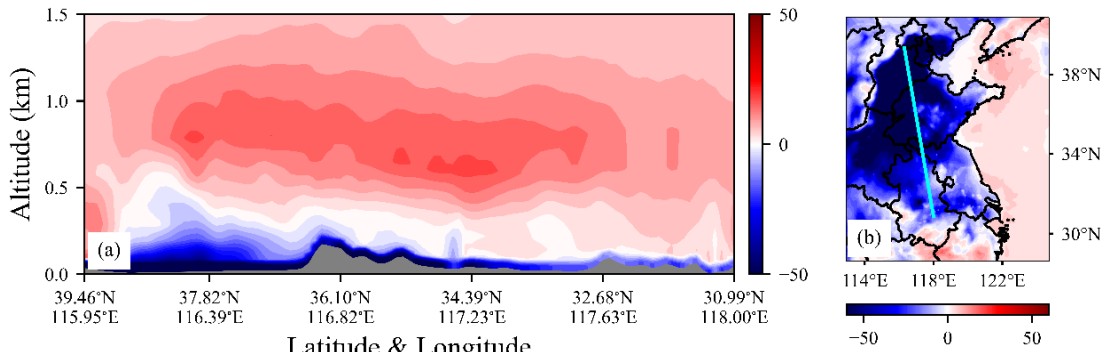


**Figure 3.** The distribution of difference (EXP_NEW-EXP_BASE) in PM$_{2.5}$ concentration (unit: μg·m$^{-3}$). (a) Vertical cross section of the difference (blue line in Figure 3b). (b) Difference on the surface.

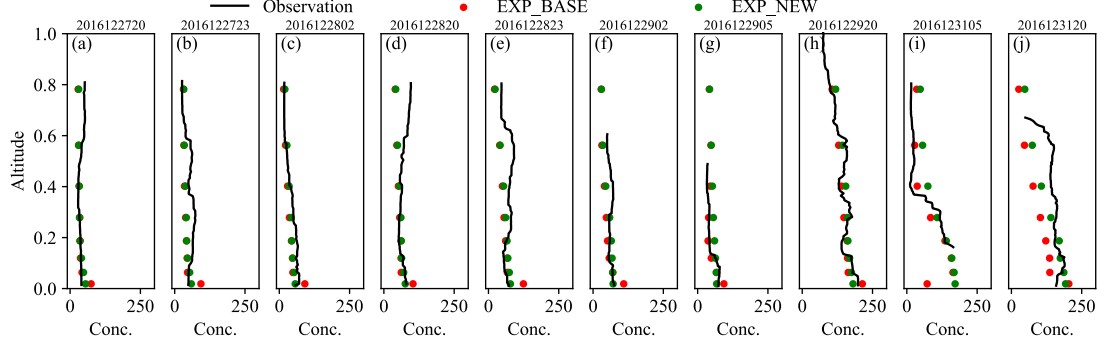

**Figure 4.** Model performance of PM$_{2.5}$ (unit: ug·m$^{-3}$) in vertical direction. The black solid line represents the observation.

The red and grey dot represents the simulation in EXP_BASE and EXP_NEW, respectively. Time in an orange shading represents nighttime.




## 3.2 The relationship between Kzmin and the evaporative fraction

As shown in Figure 5, the new Kzmin scheme enhanced the turbulent diffusion values over eastern China, much larger than the default value of 0.01 $m^2 \cdot s^{-1}$. The distribution of the Kzmin value exhibited a north–south difference with a mean

value of 0.88 $m^2 \cdot s^{-1}$ (0.8-1.3 $m^2 \cdot s^{-1}$) in the NCP and 1.17 $m^2 \cdot s^{-1}$ (1.0-1.5 $m^2 \cdot s^{-1}$) in the YRD, which is within the ideal Kzmin range based on the sensitivity experiments in section 2.3 (Table S1).

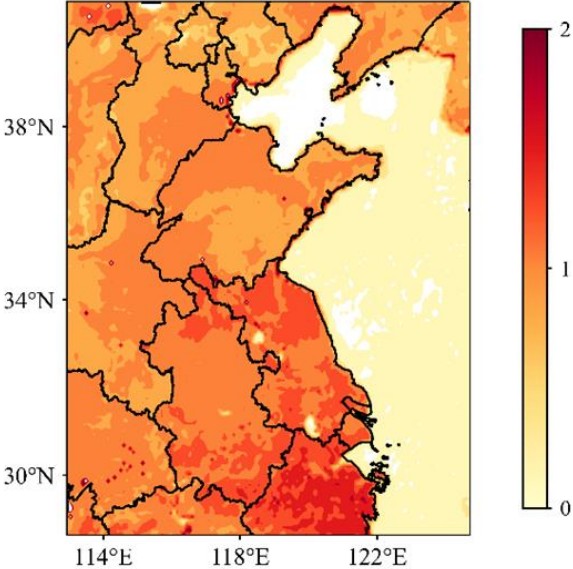

**Figure 5.** The distribution of Kzmin (unit: $m^2 \cdot s^{-1}$) in EXP_NEW.


EF in formula 4 was used to represent the proportion of available energy at the surface that is used for evaporation (i.e., LE) (Shuttleworth et al., 1989). A high value of EF indicated that a large portion of the available energy was being used for evaporation, which is associated with more stable atmospheric conditions. The evaporation process tends to absorb heat and cool the surface. In contrast, a low value of EF indicated that a larger proportion of the available energy at the surface is used

to heat the air directly through H, leading to a higher air temperature near the surface and reducing the stability of the atmosphere. We need to use a larger Kzmin value to enhance turbulence under a strong stable atmosphere and small or no adjusted Kzmin values under a weak stable or neutral atmosphere. As such, EF can reflect thermal flux features related to climate and the underlying surface in different regions, e.g., the NCP and YRD. After setting the adjustment factor value to 1.0 in formula 3, we obtained reasonable Kzmin values suitable for $PM_{2.5}$ simulation under the SBL over East China.



## 3.3 Key process in the improvement of PM$_{2.5}$ simulation in the new scheme


As shown in section 3.1, the average PM$_{2.5}$ concentration during nighttime in EXP_NEW was significantly reduced, and the values were much closer to the observed data. To further investigate the key processes that contributed to the improvement of PM$_{2.5}$ in EXP_NEW, we analysed the changes in PM$_{2.5}$ induced by each physical and chemical process in the simulations. There was a significant difference in advection (ADV), vertical mixing (VMIX), and aerosol process

(AERO) between the two schemes.

VMIX is related to turbulent diffusion in the vertical direction, which is important for the mixing and transport of air pollutants in the vertical direction. As most primary pollutants are emitted to the first level in the model, the PM$_{2.5}$ concentration at the surface is typically higher than that at higher altitudes. As a result, PM$_{2.5}$ is mixed from the surface to higher altitudes via turbulent diffusion, leading to negative VMIX values at the surface in most regions (Figure 6a, b). The

difference in VMIX contribution (Figure 6c) indicated that the new scheme improved the overestimation of PM$_{2.5}$ near the surface. More surface PM$_{2.5}$ can diffuse to high altitudes in EXP_NEW, which means that VMIX reduces the underestimation of PM$_{2.5}$ in the upper BL (discussed in Figure 7a). Regarding the advection process (ADV), there are no significant distribution characteristics between EXP_BASE and EXP_NEW (Figure 5d, 5e). Wind can transport pollutants among different regions or play a role in clearing air pollutants (Kang et al., 2019). Figure 5f shows that the difference

between EXP_NEW and EXP_BASE is positive in most regions of eastern China. The ADV process in EXP_NEW increased the surface PM$_{2.5}$ concentration compared with that in EXP_BASE. Therefore, the ADV process is not the key process in improving overestimation issues. It only redistributed the PM$_{2.5}$ over this region. AERO mainly involves the conversion of aerosol and precursor gases. The simulation temperature in EXP_NEW was higher than that in EXP_BASE (Table 1) near the surface, which may accelerate the production of PM$_{2.5}$. The positive difference between AERO in

EXP_NEW and EXP_BASE (Figure 6i), especially in the YRD region, indicates that the AERO process did not contribute significantly to the improvement of surface PM$_{2.5}$ simulation.



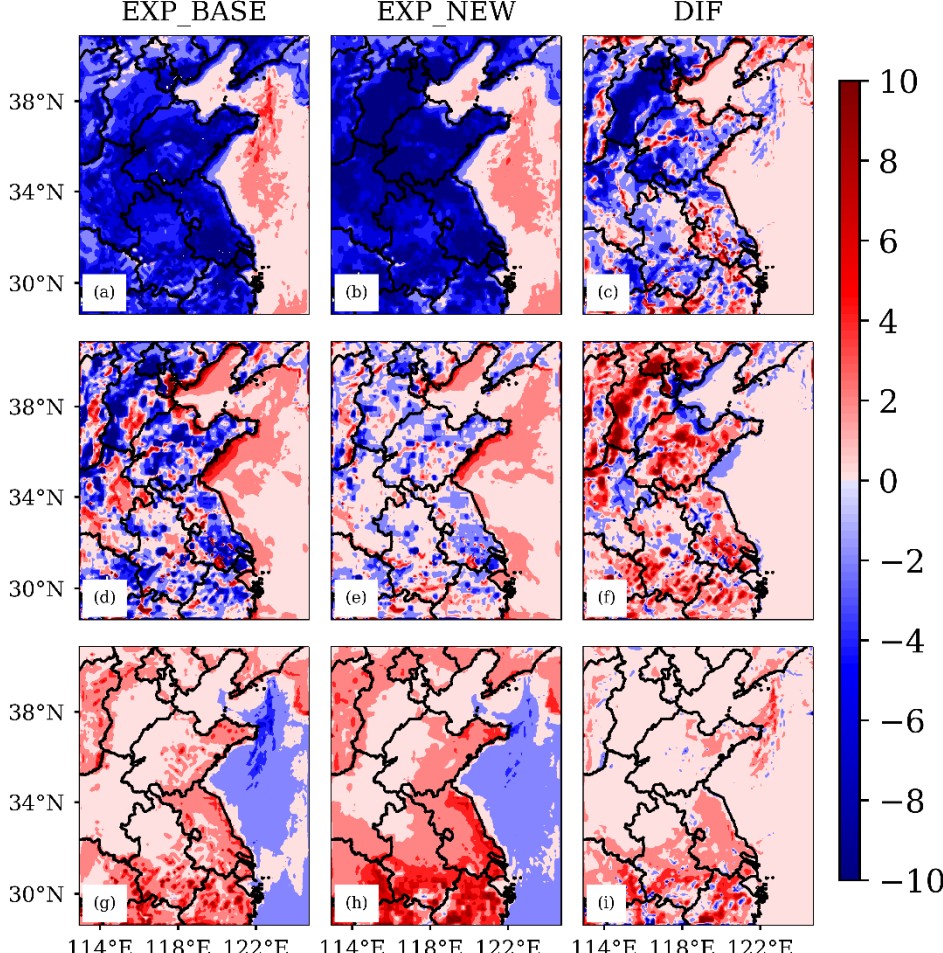

**Figure 6.** The distribution of PM$_{2.5}$ contributions from VMIX (unit: ug·m$^{-3}$·h$^{-1}$), ADV (unit: ug·m$^{-3}$·h$^{-1}$), and AERO (unit: ug·m$^{-3}$·h$^{-1}$) in EXP_BASE (a, d, g) and EXP_NEW (b, e, h). (c, f, i) represents the difference between EXP_NEW and EXP_BASE.

Figure 7 shows a vertical cross section of the differences from process contributions along the NCP to the YRD. The difference in VMIX is negative in the lower BL (< 0.5 km) but positive in the upper BL (>0.5 km), which means that VMIX in the new scheme enhanced turbulence diffusion and diffused more PM$_{2.5}$ from the lower BL to the upper BL. Both benefits improved the overestimation on the surface and underestimation at high altitudes. For the difference in ADV, the effect is the opposite of that of VMIX, which is positive at the lower BL and negative at the upper BL. This result indicated that ADV increased the PM$_{2.5}$ concentration at the lower BL and decreased it at the upper BL. The difference in AERO is positive from the surface to the upper BL. More precursors can be diffused by turbulence in EXP_NEW, which could contribute to the formation of secondary aerosols, such as nitrate. The net contribution showed that the combined effect of various processes in EXP_NEW decreased the surface concentration and increased the upper BL concentration. The decrease effect on the




surface can contribute to the VMIX. The increase effect between 0.1-0.5 km could contribute to the ADV and AERO, and the increase effect above 0.5 km may contribute to the VMIX and AERO.



**Figure 7.** The distribution of difference (EXP_NEW-EXP_BASE, unit: ug·m⁻³·h⁻¹) in difference process contribution (a: VMIX, b: ADV, c: AERO d: NET=VMIX+ADV+AERO).




Through process analysis, we discovered that the VMIX process decreased the surface PM$_{2.5}$ concentration in the nocturnal SBL, while both the AERO and ADV processes had the opposite effect. The results suggest that our new parameterized Kzmin scheme improved PM$_{2.5}$ simulation mainly driven by the VMIX process on the surface. The improvement in upper-BL was related to the combined effects of the three processes.

## 4 Conclusion

Accurately simulating PM$_{2.5}$ under the stable boundary layer (SBL) remains a challenging issue in model studies, as weak turbulence often results in overestimation of PM$_{2.5}$ concentrations. The default Kzmin value (0.01 m$^2$·s$^{-1}$) in WRF-Chem 3.9.1 is too small to facilitate nocturnal turbulence diffusion. In this study, the new parameterization of Kzmin was applied and enhanced the PM$_{2.5}$ simulation performance in the nocturnal SBL over eastern China. Furthermore, we discussed the physical relationships of the parameterized formula and explored the key process in the improvement of simulation. The conclusions are described as follows:

Sensitivity experiments show that there are different value ranges of available Kzmin over the northern (0.8 to 1.3 m$^2$·s$^{-1}$) and southern (1.0 to 1.5 m$^2$·s$^{-1}$) regions of East China. Using a fixed Kzmin may present a flaw. We determined that the geographically related Kzmin could be parameterized by the evaporative fraction (EF), which also exhibited a regional difference related to the climate and underlying surface. Thus, we parameterized Kzmin to enhance the turbulent diffusion in the SBL and embedded it into the YSU scheme. The model overestimated (98.8%) the simulation of PM$_{2.5}$ with a mean relative bias equal to 43.0% in EXP_BASE, where Kzmin is equal to 0.01 m$^2$·s$^{-1}$. The mean value of MB is 54.61 µg·m$^{-3}$ in the NCP and 37.05 µg·m$^{-3}$ in the YRD. Compared with EXP_BASE, EXP_NEW, where Kzmin is parameterized, can significantly improve the model simulations of temperature and PM$_{2.5}$. The issue of overestimated surface PM$_{2.5}$ under the SBL has been solved, with the mean relative bias decreasing to 15.6%. The mean value of MB decreases to 3.79 µg·m$^{-3}$ in the NCP and 17.99 µg·m$^{-3}$ in the YRD, which is more noticeable in the north. EXP_NEW also improved the underestimation of PM$_{2.5}$ in the upper SBL, while the improvement was not obvious compared to the surface.

We also determined the relationship between Kzmin and EF in the SBL that we used in the parameterized formula. The value of the evaporative fraction, which was calculated by H and LE, can represent the proportion of available energy at the surface that is used for evaporation. A high value of EF was consistent with a more stable atmosphere, and a larger Kzmin was needed. Therefore, we parameterized Kzmin with EF, which is calculated by H and LE. In this process analysis, VMIX is the key process for the improvement of PM$_{2.5}$ simulation on the surface in EXP_NEW, which made a negative contribution to the surface PM$_{2.5}$ concentration. AERO and ADV both increased the surface PM$_{2.5}$ concentration in the SBL, thereby having a counterproductive effect on the simulation improvement. The increase in PM$_{2.5}$ concentration in the upper SBL was attributed to VMIX, ADV and AERO. We highlight the importance of enhanced turbulent diffusion in the current mesoscale models under the SBL to improve the PM$_{2.5}$ simulation.



**Code and data availability.** The ground air quality data are from the China National Environmental Monitoring Center and
published online (https://air.cnemc.cn:18007/, last accessed: May 18, 2023). The WRF-Chem source code is from
https://www2.mmm.ucar.edu/wrf/users/download/get_source.html. The model output can contract to Bin Zhu
(binzhu@nuist.edu.cn). The observation data, and modified codes in our work are available online via Zenodo
(https://doi.org/10.5281/zenodo.8016692, Lu ,2023).

**Author contributions.** WL performed the model simulation, data analysis, and writing of the paper. BZ proposed the idea,
supervised this work, and revised the paper. SY, JL and ZW offered to help with the model simulation and the revision of the
paper.

**Competing interests.** The contact author has declared that neither they nor their co-authors have any competing interests.

**Acknowledgements.** The numerical calculations in this paper have been done on the high-performance computing system in
the High-Performance Computing Center, Nanjing University of Information Science and Technology.

**Financial support.** This work is supported by the National Natural Science Foundation of China (Grant Nos. 92044302 and
42275115) and the Postgraduate Research and Practice Innovation of Jiangsu Province Program (Grant No. KYCX20_0952).

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
