# Peer review of "Parameterized minimum eddy diffusivity in WRF-Chem(v3.9.1.1) for improving $PM_{2.5}$ simulation in the stable boundary layer over eastern China"

_EGUsphere, 2023_

## Author Comment (AC1)

Dear editor and reviewers,

Thank you for giving us the opportunity to revise our manuscript (Manuscript Number: egusphere-2023-1089). We appreciate your constructive comments and detail suggestions, which we have studied carefully while making appropriate revisions on the manuscript. We believe that under your guidance, our manuscript has been substantially improved.

In the following, the reviewer's comments/suggestions are highlighted by gray. The symbol "≫" quotes the original texts in the manuscript. Followed by the comments are our responses in plain text, as well as the respective revisions in the manuscript. Some important revisions are marked with red font.

Thank you again for your constructive comments and suggestions.
Yours sincerely,

Wen Lu, Bin Zhu, and all co-authors.

**Replies to Reviewer**

This article proposes a parametrization for the diagnostic of a minimum value of eddy diffusivity. The goal is to improve the modelling of PM$_{2.5}$ in the stable boundary layer. The scheme is applied to a test case in China with the WRF-Chem regional meteorological model. They diagnosed that model generally underestimates weak turbulence in the nocturnal stable boundary layer leading to an overestimation of surface aerosol concentrations. To parameterize this Kzmin, they propose to use the sensible H and latent LE heat fluxes. The additional term increases the Kzmin value and enables to reduce significantly their model bias. They show that their change includes a spatial variability of the Kzmin, depending on the landuse and the meteorology. A latitudinal effect is diagnosed.

Thank you for investing the time to review our paper. We are very glad to receive your pertinent proposals. We think your suggestions and comments have greatly improved our manuscript in science and technical details.

**General Comments**
**Comment 1:**
The fact to consider that all models overestimate surface concentrations of particles is not correct. It is the specific case of this model WRF-chem. But there is no systematic tendency about this point. If the authors are sure of that, please provide the bibliography, a review article.

Thank you for this valuable suggestion. The statement that all models overestimate surface PM$_{2.5}$ concentration in stable boundary layer (SBL) should be arbitrary from a global view, which only related to the specified region for the most cases, eg. in eastern China. We did not find a systematic review article which have evaluated model performance of particles in SBL in eastern China. However, many scholars have reported the overestimation of PM$_{2.5}$ in eastern China, especially in SBL condition (WRF-Chem: Du et al., 2020, Jia et al., 2021, Qiu et al., Wang et al., 2021; WRF-NAQPMS: Chen, 2022; WRF-CMAQ: Liu et al., 2023). The meteorological conditions for these models are all provided by the WRF model. Therefore, here, we focused the common overestimates of the particle concentration in SBL in eastern China by WRF provided meteorological conditions, which could be related to the incorrectly simulation of meteorological factors in SBL. In addition, we admit the reviewer's

point and found that there were some underestimates in other regions. For example, Zhang et al. (2020) used GEOS-Chem, WRF-Chem, and CMAQ to evaluate the model performance of $PM_{2.5}$ in north America. They found that all CTMs underestimates monthly mean $PM_{2.5}$ concentration compared with ground observations. In the conclusion, we add the potential uncertainty of $PM_{2.5}$ simulations in our research in the revised manuscript of line 312 to 317.

**References and their related presentation:**

[1] Chen (2022). The numerical simulation of critical control process of aerosol vertical structure. [D]. Beijing: Institute of Atmospheric Physics, Chinese Academy of Sciences, 63-66.

[2] Du, Q., Zhao, C., Zhang, M., Dong, X., Chen, Y., Liu, Z., ... & Miao, S. (2020). Modeling diurnal variation of surface $PM_{2.5}$ concentrations over East China with WRF-Chem: Impacts from boundary-layer mixing and anthropogenic emission. Atmospheric Chemistry and Physics, 20(5), 2839-2863.

*In page 7: the CTL1 experiment can generally capture the diurnal variation of the DI of surface $PM_{2.5}$ in the four cities, but overestimates the DI in the night, particularly in spring and autumn.*

[3] Jia, W., Zhang, X., Zhang, H., & Ren, Y. (2021). Application of turbulent diffusion term of aerosols in mesoscale model. Geophysical Research Letters, 48(11), e2021GL093199.

*In page 4: In short, the overestimation of pollutant concentration is still a common problem, and it is worthy of research and discussion.*

[4] Qiu, Y., Liao, H., Zhang, R., & Hu, J. (2017). Simulated impacts of direct radiative effects of scattering and absorbing aerosols on surface layer aerosol concentrations in China during a heavily polluted event in February 2014. Journal of Geophysical Research: Atmospheres, 122(11), 5955-5975.

*In abstract: Comparisons of model results with observations showed that the WRF-Chem model reproduced the spatial and temporal variations of meteorological variables reasonably well but overestimated average PM2.5 concentration by 21.7% over the NCP during 21–27 February.*

[5] Liu, M., Lin, J., Wang, Y., Sun, Y., Zheng, B., Shao, J., ... & Wu, Z. (2018). Spatiotemporal variability of $NO_2$ and $PM_{2.5}$ over Eastern China: observational and model analyses with a novel statistical method. Atmospheric Chemistry and Physics, 18(17), 12933-12952.

*In abstract: CMAQ overestimates the diurnal cycle of pollutants due to too-weak boundary layer mixing, especially in the nighttime, and overestimates $NO_2$ by about 30 ug·m$^{-3}$ and $PM_{2.5}$ by 60 ug·m$^{-3}$.*

[6] Zhang, H., Wang, J., García, L. C., Ge, C., Plessel, T., Szykman, J., ... & Spero, T. L. (2020). Improving surface $PM_{2.5}$ forecasts in the United States using an ensemble of chemical transport model outputs: 1. Bias correction with surface observations in nonrural areas. Journal of Geophysical Research: Atmospheres, 125(14), e2019JD032293.

*In abscract: While all CTMs (CMAQ, WRF-Chem, GEOS-Chem) underestimate daily surface $PM_{2.5}$ mass concentration by 20–50%, KF correction is effective for improving each CTM forecast.*

**In the last of our revised manuscript in Line 312-317:**

It is worth noting that is one kind of error compensation for enhance the underestimated turbulent diffusion under the stable boundary layer. In the absence of effective physical scheme in thermodynamics, it is an alternative choice. Although there have been many studies reporting the overestimation within the SBL in eastern China. However, models are not always overestimated in other countries and regions. Adaptation our scheme to other regions and other models needs to be further evaluated. In this study, winter pollution in eastern China was investigated and only one month of simulation was done, and simulations for other seasons need to be further evaluated.

**Comment 2:**

But considering this is mostly a problem with this model, the fact to add a term to reduce this bias is a tuning. Except if the new term has a robust physical basis. For the moment, this additional Kzmin is able to unbias the model, but it could be only an error compensation. Of course, it is always difficult to quantify but the authors should at least discuss this point and add more sensitivity tests: injection of anthropogenic emissions at levels higher that the surface level, test of boundary layer scheme to see the model sensitivity to the bias of $T_{2m}$, among others.

Thank you for this valuable suggestion. As you said, Kzmin is one kind of error compensation for the underestimated turbulent diffusion in the SBL and is difficult to quantify in thermodynamics. Therefore, in our original manuscript, we set a series of experiments with different fixed Kzmin value to obtain the reasonable Kzmin value ranges in eastern China. As your suggestion, more experiments were tested. The first series experiments are tested the sensitivity of $PM_{2.5}$ to different inject height. In our study, the anthropogenic emission inventory that we used is MEIC (in China) and MIX (east Asia that excluding China). MEIC and MIX are divided into five sector emission, including (power, residential, transportation, residential and industry emission). Two experiments were set to compare the effects of emission inject height:

1. EXP_H7: According to the vertical emission ratio profile suggestion by MEIC, residential, transport and residential emission were emitted to the first level in the model. For power emission, it was emitted from about 61-550m (level 2 to level 7). For the industry emission, it was emitted from 18m-116m (level 1 to level 3).

2. EXP_H2: The residential, transportation, industry and residential emission were emitted into the first level in the model. For power emission, it was emitted into the second level.

The emission inject height of different sector source was shown in Figure R1 and the mean model performance of the $PM_{2.5}$ simulation was shown in Table R1. The simulation result of $PM_{2.5}$ is much better in EXP_H7, which primary pollutant was emittd to higher altitudes. However, the result in EXP_H7 was still overestimated. Our EXP_BASE experiment used the same emission inject heights as EXP_H7. Therefore, we believe we need to condisider the other ways to improve the overestimation of $PM_{2.5}$.

[Figure]

**Figure R1.** The emission percentage (%) in two experiment.

**Table R1.** Mean model performance metrics for and PM$_{2.5}$ (nighttime) in different emission inject height.

| Case_name | MB | IOA | RMSE | R |
|---|---|---|---|---|
| EXP_H7 | 48.23 | 0.72 | 82.68 | 0.66 |
| EXP_H2 | 60.34 | 0.7 | 91.11 | 0.62 |

You may be interested in the sensitivity of boundary layer scheme to the bias of T$_{2m}$. The incorrect T$_{2m}$ simulation has also received previous attention. Hu et al., (2010) has tested three boundary layers (YSU, MYJ, ACM2) to see the model sensitivity to the bias of T$_{2m}$ by WRF model. They indicated that the use of the local-closure MYJ scheme produces the largest bias. The YSU PBL scheme produces higher temperatures than with the other two schemes during nighttime in the lower atmosphere. Jia et al. (2023) found that the differences in simulated temperatures between the nonlocal scheme mainly originate from downward shortwave radiation, while the simulation differences in local closure PBL schemes may be related to the simulated difference in sensible heat flux. In general, the simulation of T$_{2m}$ still needs to be improved, and we will focus on it in future.

**Reference**

Hu, X. M., Nielsen-Gammon, J. W., & Zhang, F. (2010). Evaluation of three planetary boundary layer schemes in the WRF model. Journal of Applied Meteorology and Climatology, 49(9), 1831-1844.

Jia, W., Zhang, X., Wang, H., Wang, Y., Wang, D., Zhong, J., Zhang, W., Zhang, L., Guo, L., Lei, Y., Wang, J., Yang, Y., and Lin, Y.: Comprehensive evaluation of typical planetary boundary layer (PBL) parameterization schemes in China. Part I: Understanding expressiveness of schemes for different regions from the mechanism perspective, Geosci. Model Dev. Discuss. [preprint], https://doi.org/10.5194/gmd-2023-30, in review, 2023.

**Comment 3:**

A detailed analysis based on hourly time-series and comparison to surface observation of PM$_{2.5}$ is also missing to really see if there is a physically improvements of the surface concentrations with this scheme. Ideally, lidar data could help to see if the vertical structure of aerosols is better reproduced by the model.

Thank you for this valuable suggestion. We are apologized for not providing the hourly time-series to show our improved results. Here, we provide the results in supplement Figure S3. The simulation of four cities (2 north cities in north; 2 cities in south) was used to demonstrate the improvements. They are Zhengzhou (34.75°N, 113.64°E); Jining (35.43°N, 116.63°E) in north of China and Hefei (31.94°N, 117.27°E); Nanjing (32.2°N, 118.8°E) in YRD. There are three to four haze events (the concentration of PM$_{2.5}$ exceeds 115 ug·m$^{-3}$ and lasts for more than 48 hours, shaded in yellow) occurred in each city in December 2016. The overestimation of EXP_BASE is obvious, especially during stable condition, e.g., during nighttime, and heavy pollution events. The model result in EXP_NEW is improved by using our parameterized Kzmin and closer to the observation. The metrics such as MB and IOA has significant improvement (Figure S3). It's worth noting that both two schemes underestimate extreme high peak concentrations of PM$_{2.5}$ (such as Zhengzhou on 19 December, the concentration greater than 600 ug·m$^{-3}$), which may be due to the poor ability to extreme heave pollution simulation in existing mesoscale models. The hourly time-series was added in the revised supplement Figure S2.

For the vertical structure of PM$_{2.5}$, the field experiment data was used. The observation site is located in the northern suburbs of Nanjing. The coordinates and altitude of the observation site are 32.0°N, 118.4°E and approximately 23 m asl, respectively. The PM$_{2.5}$ concentrations were measured by a PDR-1500 fixed on an unmanned aerial vehicle (UAV) platform from December 27, 2016, to December 31, 2016. 10 profiles (surface to ~1.0km) of PM$_{2.5}$ in SBL were obtained for model evaluation in vertical. We have well evaluated the PM$_{2.5}$ observed by PDR-1500 (Zhu et al., 2019, Shi et al., 2021). The

improvement in the simulation on the surface is significant, and some underestimation periods in the high altitude is also improved. In general, the profile in EXP_NEW is closer to the profile observed in the vertical. We are sorry we cannot get high quality PM$_{2.5}$ data retrieved from lidar, because they are observed in extinction coefficient and not well evaluated in high quality from individual maintain institutions. The picture was revised in Figure 4 of the revised manuscript.

**Reference:**

Shi, S., Zhu, B., Lu, W., Yan, S., Fang, C., Liu, X., ... & Liu, C. (2021). Estimation of radiative forcing and heating rate based on vertical observation of black carbon in Nanjing, China. Science of The Total Environment, 756, 144135.

Zhu, J., Zhu, B., Huang, Y., An, J., & Xu, J. (2019). PM$_{2.5}$ vertical variation during a fog episode in a rural area of the Yangtze River Delta, China. Science of The Total Environment, 685, 555-563.

In the last of our revised supplement in Figure S2

[Figure]

**Figure S2.** Time series of PM$_{2.5}$ concentration in Zhengzhou, Jining, Hefei and Nanjing in December 2016. The grey dots, red lines and blue lines represent the results of observation, EXP_BASE and EXP_NEW, respectively. The yellow shaded represent haze events (the concentration of PM$_{2.5}$ exceeds 115 ug·m$^{-3}$ and lasts for more than 48 hours) in each city. The metrics (MB, IOA and R) was calculated by the full day (daytime and nighttime) data.

**In the last of our revised manuscript in Figure 4**

[Figure]

**Comment 4:**

Another question: the comparison is only perfomed for PM$_{2.5}$. What about PM$_{10}$? NO$_2$ and O$_3$ (often measured at stations)? The bias on these species should be of interest to understand if the new Kzmin value is really better for all species.

We tested the simulation of PM$_{10}$ and two gases (O$_3$ and NO$_2$). The mean model performance was shown in Table R2. As for PM$_{10}$, EXP_BASE overestimated the simulation, which is very similar to the PM$_{2.5}$. The model performences of EXP_NEW was better than EXP_BASE with the smaller mean bias and larger IOA and R. For NO$_2$, model overestimated the simulation in EXP_BASE. EXP_NEW decreased the simulation MB from 61.12 ug·m$^{-3}$ to 34.95 ug·m$^{-3}$ and increase IOA from 0.44 to 0.52. In general, the simulation result of PM$_{10}$ and NO$_2$ in EXP_NEW are better than that in EXP_BASE. While the simulation result is poor for O$_3$ than PM$_{10}$ and NO$_2$, both in EXP_BASE and EXP_NEW, which may be related to the poor ability to simulate nighttime ozone in WRF-Chem model. Delightfully, EXP_NEW decrease the mean bias from -19.49 ug·m$^{-3}$ to -12.15 ug·m$^{-3}$. We will focus on the improvement of nighttime O$_3$ simulation in the future.

**Table R2.** Mean model performance metrics for PM$_{2.5}$, O$_3$, and NO$_2$ (nighttime).

| Species name | Case_name | MB | IOA | RMSE | R |
|---|---|---|---|---|---|
| PM$_{10}$ | EXP_BASE | 28.72 | 0.79 | 73.73 | 0.73 |
| | EXP_NEW | -6.85 | 0.83 | 57.31 | 0.75 |
| NO$_2$ | EXP_BASE | 61.12 | 0.44 | 77.5 | 0.54 |
| | EXP_NEW | 34.95 | 0.52 | 51.84 | 0.47 |
| O3 | EXP_BASE | -19.49 | 0.5 | 27.22 | 0.27 |
| | EXP_NEW | -12.15 | 0.5 | 27.19 | 0.2 |

**Specefic comments:**

**Comment 1:**

The abstract is clear and summarizes correctly the whole content of the study, altough the last three sentences deserved to be reformulated.

≫ Line31-35: … Process analysis showed that vertical mixing is the key process to improve $PM_{2.5}$ simulations on the surface in the revised scheme. The increase in the $PM_{2.5}$ concentration in the upper SBL was attributed to vertical mixing, advection, and aerosol chemistry. This study highlights the importance of improving turbulent diffusion in current mesoscale models under the SBL and has great significance for aerosol simulation research under heavy air pollution events.

Thank you for this valuable suggestion. The last three sentences have been reformulated in revised manuscript of line 31-34.

**In the last of our revised manuscript in Line 31-34:**

… Process analysis showed that vertical mixing is the key process to improve the overestimation of surface $PM_{2.5}$ simulation under the SBL. This study suggests that a stronger turbulent diffusion is required in current mesoscale models to better simulate the surface $PM_{2.5}$ under the SBL.

**Comment 2:**

Data and Metodology: Several data are used to validate the model's hypotheses made in this study. It includes vertical measurements, essential for this type of vertical mixing study. The model used is WRFchem, known as a fully coupled model. Unfortunately, the coupling is not always really activated, all options being not coupled. It is recommended to the authors to add in Appendix an explanation about their namelist to ensure that the coupling was really fully active. The choice in the namelist may completely change their results.

Thanks for your concerns. We are sure that the combination of our schemes is fully coupled. The aerosol feedback is already enabled in the namelist.input file. The detail of the physical and chemical parameterization schemes was added in the revised spplement Table S2 and the namelist.input file was provided in the supplement.

**In the last of our revised manuscript in**

**Line 89-90:**

… Other physical and chemical parameterization schemes was shown in Table S2.

**In the last of our revised supplement in**

**Table S2.** Physical and chemical parameterization schemes.

| Scheme | Option |
|---|---|
| Boundary layer | YSU (Hong et al., 2006) |
| Microphysics | Morrison (Morrison et al., 2009) |
| Longwave radiation | RRTMG (Iacono et al., 2008) |
| Land surface | Noah (Chen et al., 2001) |
| Gas-phase chemistry | CBM-Z (Zaveri et al., 1996) |
| Aerosol chemistry | MOSAIC-8bin (Zaveri et al., 2008) |
| Aerosol–radiation feedback | On |

**Comment 3:**

The key point of the study is to assume that the 'evaporative fraction' (EF) may be used to characterize the searched value of Kzmin. Why not, but why exactly?

As shown in general comment 1, the overestimation of PM$_{2.5}$ concentration under the SBL is common in eastern China which related to the underestimation of turbulance diffusion intensity. The poor simulated results of turbulent diffusion in SBL may be related to laking of the full conceptual and theoretical understanding in SBL. Kzmin has no physical significance in YSU scheme of the original model, but is one kind of simply an error compensation for the lack of turbulent diffusion capacity for enhance the underestimated turbulent diffusion under the SBL. In the absence of effective physical scheme in thermodynamics, it is an alternative choice. Our work is to raise a reasonable dynamic Kzmin and appliy it in eastern China for improvement the PM$_{2.5}$ simulation under SBL. In the multiple fixed Kzmin value experiments (Table S1), the reasonable Kzmin value ranges was obtained for the north China plain (NCP) (0.8 to 1.3 m$^2$·s$^{-1}$) and YRD region (1.0 to 1.5 m$^2$·s$^{-1}$) of eastern China. Compared to the north region, the YRD needs a larger Kzmin value for PM$_{2.5}$ simulation at SBL. Fortunately, EF can reflect the differences of physical characteristics between NCP and YRD, which related to the meteorological features of radiation, temprature, cloud, precipitation, underlying surface (soil moisture, L,H), etc. in different climate zones (Han et al., 2020, Jin et al., 2021). Also, we found the value of EF+1.0 was consistent with the distribution of the reasonable Kzmin value ranges in north and south regions of eastern China, showing in sensitive experiments (Table S1). Therefore, we intend to use EF+1.0 to parameterize the value of Kzmin for improving PM$_{2.5}$ simulation over eastern China and the improvement was obvious for PM$_{2.5}$ simulation in Section 3.1.

**Reference**

Han, G., Wang, J., Pan, Y., Huang, N., Zhang, Z., Peng, R., ... & Pan, Z. (2020). Temporal and spatial variation of soil moisture and its possible impact on regional air temperature in China. Water, 12(6), 1807.

Jin, H., Chen, X., Wu, P., Song, C., & Xia, W. (2021). Evaluation of spatial-temporal distribution of precipitation in mainland China by statistic and clustering methods. Atmospheric Research, 262, 105772.

**Comment 4:**

The end of the paragraph (lines 130 to 137) is not very clear and should be reworded. It means that under stable conditions, the values of Kzmin may be 100 times larger than under unstable conditions?
≫ Line130-137: … and the expression can be found in formula 1.

$$under\ stable \quad : Kzmin = EF + 1.0 \tag{5}$$

$$under\ unstable : Kzmin = 0.01 \tag{6}$$

When the grid in the PBL was under stable conditions (Ri > 0), the Kzmin value was set to the value calculated by formula 1. While the grid in the PBL was under unstable conditions (Ri<0), the Kzmin value was set to the default value (0.01). To avoid outlier calculation results, we set the Kzmin value variations from 0.01 to 2.0 (93% grid values fall within this interval). By comparing EXP_BASE with EXP_NEW, we can explore the impact of Kzmin on the PM$_{2.5}$ simulation. We will also discuss the physical relationships of Kzmin with EF in section 3.2.

The manuscript may contain unclear descriptions in this paragraph. The turbulent diffusion coefficient K$_h$ can be calculated following fomula 3: $K_h = K_m/P_r + Kzmin$, where K$_m$ is momentum mixing coefficient, Pr is the prandtl number.

The default value of the Kzmin in YSU PBL scheme is 0.01 m$^2$·s$^{-1}$. The aim of our work is to improve the simulation for the stable condition. Therefore, we only parameterized the Kzmin value during stable condition. While for the unstable, we let Kzmin consistent with the default setting (0.01 m$^2$·s$^{-1}$). In average, we find that the new adjusted K$_h$ were 1.35 m$^2$·s$^{-1}$ in NCP

and 2.03 $m^2 \cdot s^{-1}$ in YRD, which are much lower than the $K_h$ in daytime (bigger than 3 $m^2 \cdot s^{-1}$). For example, the values of $K_h$ in EXP_BASE and EXP_NEW in Nanjing (32.0°N,118.4°E) was shown in Figure R2. There is no $K_h$ value at stable condition larger than that in unstable condition. We revised the text in line 134-138 of revised manuscript.

[Figure]

**Figure R2.** The turbulent diffusion coefficient in EXP_NEW and EPX_BASE (unit: $m^2 \cdot s^{-1}$) in Nanjing. The yellow shaded represent the stable condition (Richardson number>0).

**In the last of our revised manuscript**

**Line 134-138**

… As such, we parameterized a new value of Kzmin in the PBL scheme in SBL based on the results of the sensitivity experiments that in EXP_NEW, and the expression can be found in formula 5:

$$Kzmin = EF + 1.0 \tag{5}$$

When the grid was under stable conditions (Ri > 0), the Kzmin value was set to the value calculated by formula 5. …

**Comment 5:**

If some metrics are well known, please define them, including the IOA Index Of Agreement.

Thank you for this valuable suggestion. The definition of the metrics that we used was added in the revised supplement of line 10 to 17.

**In the last of our revised supplement:**

**Line 10-17:**

Mean Bias: $MB = \frac{1}{N}\sum_{i=1}^{N}(M_i - O_i)$

Index Of Agreement: $IOA = 1 - \frac{\sum_{i=1}^{N}(M_i-O_i)^2}{\sum_{i=1}^{N}(|M_i-\bar{M}|+|O_i-\bar{O}|)^2}$

Root Mean Square Error: $RMSE = \sqrt{\frac{1}{N}\sum_{i=1}^{N}(M_i - O_i)^2}$

Correlation Coefficient: $R = \frac{Cov(x,y)}{\sqrt{x}\sqrt{y}}$

Normalized Mean Bias: $NMB = \frac{\sum_{i=1}^{N}(M_i-O_i)}{\sum_{i=1}^{N}O_i} \times 100\%$

Normalized Mean Error: $NME = \frac{\sum_{i=1}^{N}|M_i-O_i|}{\sum_{i=1}^{N}O_i} \times 100\%$

**Comment 6:**

l.145: if the key point is an enhancement of Kzmin during the night, why not show time series of a few days, at a station with measurements and with hourly values, showing three to four consecutive days?

Thank you for this valuable suggestion. The simulation of four citys and the station information can be found in the answer of comment 3. There are three to four haze events (the concentration of $PM_{2.5}$ exceeds 115 ug·m$^{-3}$ and lasts for more than 48 hours) occurred in each city in December 2016. We found the new scheme obviously improve the overestimated of $PM_{2.5}$. The metrics such as MB and IOA has significant improvement in EXP_NEW, especially the mean bia (MB). The value of MB decreased from 59.68 to 5.33 ug·m$^{-3}$ for Zhengzhou, from 81.54 to 13.32 ug·m$^{-3}$ for Jining, from 58.11 to 37.27 ug·m$^{-3}$ for Hefei, from 36.46 to 24.39 ug·m$^{-3}$ for Nanjing, respectively.

[Figure]

**Figure S2.** Time series of $PM_{2.5}$ concentration in Zhengzhou, Jining, Hefei and Nanjing in December 2016. The grey dots, red lines and blue lines represent the results of observation, EXP_BASE and EXP_NEW, respectively. The yellow shaded represent haze events (the concentration of $PM_{2.5}$ exceeds 115 ug·m$^{-3}$ and lasts for more than 48 hours) in each city. The metrics (MB, IOA and R) was calculated by the full day (daytime and nighttime) data.

**Comment 7:**

The Table shows that the bias for $T_{2m}$ is -1.15 degrees when the text says -0.86 degrees. Please correct the correspondence between text and tables. And a bias of 1.15 degrees is not really good. There is perhaps a problem with the use of the WRF model, independently of the Kzmin parameterization studied in this paper.

≫Line 147: … the MB mean value to be negative (-0.86 ℃) …

Thank you for this valuable suggestion. We apologize that there are some mistake in the manuscript. The mean bias for $T_{2m}$ is -1.15℃, which is not very good in the original model. While in our revised scheme, the mean bias have improved with a smaller MB value of $T_{2m}$   (0.39℃?). We have revised the text in the revised manuscript of line 152.

The incorrect temperature simulations has also received scholarly attention. Chaouch et al. (2017) found a cold bias in the 2 m air temperature during the PBL collapse and at nighttime, reflecting an overestimation of the surface cooling rate. Udina et al. (2016) suggested that WRF-LES model calculated thermal coupling at the surface is unrealistically large. As a result, the rate difference between the molecular thermal conduction and the vertical eddy diffusion is underestimated, leading to the prediction of a lower air temperature near the cooling surface in simulations. It also leads to the formation of a more stable

boundary layer compared to the observations. In general, the simulation of temperature deserves further improvement.

**Reference:**

Chaouch, N., Temimi, M., Weston, M., & Ghedira, H. (2017). Sensitivity of the meteorological model WRF-ARW to planetary boundary layer schemes during fog conditions in a coastal arid region. Atmospheric Research, 187, 106-127.

Udina, M., Sun, J., Kosović, B., & Soler, M. R. (2016). Exploring vertical turbulence structure in neutrally and stably stratified flows using the weather research and forecasting–large-eddy simulation (WRF–LES) model. Boundary-layer meteorology, 161, 355-374.

**In the last of our revised manuscript in**

**Line 152:**

… the MB mean value to be negative (-1.15°C) …

**Comment 8:**

l.165: the authors diagnosed a positive bias in $PM_{2.5}$ surface concentrations and conclude it is due to "geographical conditions, climate and emissions differences and the degree of pollution". It is not an in-depth analysis. Before tuning one parameter, it should be useful to erform some sensivity tests in order to see if the bias is due to emissions, meteorology, transport, deposition of a mixing of all.

Thank you for this valuable suggestion. As you said, it is not an in-depth analysis that the positive bias in $PM_{2.5}$ surface concentrations is due to "geographical conditions, climate and emissions differences and the degree of pollution". The NCP is located in the temperate monsoon climate and is the largest plain area in China with drier soils. The YRD is located in the subtropical monsoon zone, which has a more complex topography and wetter soils (Han et al., 2020). There are also differences in the sources of sector emission in the two regions. Lu et al., (2023) study of the sectoral black carbon primary emissions in winter reveals that most of the BC in the North China Plain comes from residential sources, while the proportion of transportation and industrial emissions in YRD were comparable to the proportion of residential sources. We try to prove our claims as much as possible through sensitivity experiments. The effects of emission inject height was tested in theresponse to general comment 2. As shown in Tables R2, the relative bias of $PM_{2.5}$ between two region was different regardless of the emission inject height. Another set of sensitivity experiments was meteorology input dataset. The input meteorology were FNL (EXP_BASE) and ERA5 (EXP_ERA5). The simulations also show north-south differences. For nightime or haze pollution, the atmosphere is usually stable and pollutant mainly comes from local emissions and the contribution of transport is smaller comapred to vertical diffusion by our simulations. As such, we argue that transport is not primary responsible for the simulated $PM_{2.5}$ overestimation under stable conditions. For deposition, no previous study has reported that the model has significant bias in this module. In general, we prefer to attribute north-south modeling differences to meteorological(or climate) and geographic conditions and emission differences. Futher sensitive experiments in meterologics, emission and deposition are necessary in future studies.

Table R1. Mean model performance metrics for and $PM_{2.5}$ (nighttime) in different experiment

| Case_name | Region | MB | IOA | RMSE | R |
|---|---|---|---|---|---|
| EXP_BASE | NCP | 57.99 | 0.76 | 89.32 | 0.73 |
| | YRD | 37.77 | 0.71 | 69.07 | 0.71 |
| EXP_height_2 | NCP | 71.28 | 0.72 | 101.15 | 0.72 |
| | YRD | 49.16 | 0.67 | 80.86 | 0.71 |
| EXP_ERA5 | NCP | 45.41 | 0.72 | 89.33 | 0.62 |
| | YRD | 29.42 | 0.73 | 62.11 | 0.66 |

**Reference**

Han, G., Wang, J., Pan, Y., Huang, N., Zhang, Z., Peng, R., ... & Pan, Z. (2020). Temporal and spatial variation of soil moisture and its possible impact on regional air temperature in China. Water, 12(6), 1807.

Lu, W., Zhu, B., Liu, X., Dai, M., Shi, S., Gao, J., & Yan, S. (2023). The influence of regional transport on the three-dimensional distributions of black carbon and its sources over eastern China. Atmospheric Environment, 297, 119585.

**Comment 9:**

l.177: What new PBL scheme? The new Kzmin formulation? or something else? This sentence seems out of place in the text.

≫Line 177: A new PBL scheme was introduced in EXP_NEW and solved the overestimation in eastern China. …

Thank you for this valuable suggestion. The PBL scheme is the YSU PBL scheme contain our new parameterized Kzmin. We have replaced it in the revised manuscript of line 183 to 184.

**In the last of our revised manuscript:**

**Line 183-184:**

The revised dynamic Kzmin parameterization was introduced into YSU PBL scheme to solve the overestimation of $PM_{2.5}$ simulation in eastern China. …

**Comment 10:**

l.187: There is no quantified improvement but "we believe that the simulation in the YRD has also been improved." Please explain better.

≫Line 186-188: … Although there is no significant improvement in the mean MB in the YRD, the simulated trend is more similar to the observation. Therefore, we believe that the simulation in the YRD has also been improved. …

Thanks. The significant imporvement is NCP relative to YRD (Figure S2). We revised the expression in the revised manuscript of line 193-194.

**In the last of our revised manuscript:**

**Line 193-194:**

… Overall, the revised scheme shows enhanced simulation results in both two regions. While the improvement in the NCP is slightly more significant compared to that in the YRD. …

**Comment 11:**

Figure 4: Usually, measurements are with symbols and model outputs with lines. Profiles are very small and difficult to read.

Thank you for this valuable suggestion. In this study, The $PM_{2.5}$ concentrations were measured by an PDR-1500 fixed on an unmanned aerial vehicle (UAV) platform (about 10-20m resolution). The observation data is denser below 800m while there are 8 grids in model outputs bellow 800m. Therefore, we use the following symbols that simulated data is dot and observed data is line. The layer with the green dot is above the layer with the red dot. The red dot in the high altitude is due to the difference is not significant between two schemes. To make the image clearer, we increased the resolution of the image and redrew the Figure 4 in the revised manuscript.

**In the last of our revised manuscript**

**Figure 4**

[Figure]

**Figure 4.** Model performance of PM$_{2.5}$ (unit: ug·m$^{-3}$) in vertical direction. The black solid line represents the observation. The red and grey dot represents the simulation in EXP_BASE and EXP_NEW, respectively.

**Comment 12:**

The choice to have a maximum possible value of 2.0 is not a result but just an arbitrary threshold choice (l.135). Then on the map in Figure 5, some values may be larger than 2.0 (7% of the values).

≫Line 135-136: To avoid outlier calculation results, we set the Kzmin value variations from 0.01 to 2.0 (93% grid values fall within this interval).

Thank you for this valuable suggestion. The setting of upper limit value of Kzmin was refer to the distribution of Kmzin which is calculate by fomula 5. We found that 2.3% of the Kzmin values were small than the default lower limit value (0.01); 93.9% of the Kzmin values were in the range of 0.01 to 2.0; 1.9% between 2.0 and 3.0; and only 1.9% lager than 3.0. Compared to the 0.01 to 2.0 interval (93.9%), only 1.9% of the Kzmin value were in the 2.0 to 3.0 interval. Table R2 give the model performaces when upper limit of Kzmin set as 1.5, 2.0 and 3.0. We can find the model peformance improve obviously when the Kzmin_uplimit change from 1.5 to 2.0, while the improvement is not significant from 2.0 to 3.0. So, we finally set the upper limit of Kzmin as 2.0.

Table R2 Mean model performance metrics for and PM$_{2.5}$ (nighttime) in different upper limit of Kzmin.

| Kzmin_upper limit | MB | IOA | RMSE | R |
|---|---|---|---|---|
| **Set to 1.5** | 17.87 | 0.77 | 63.28 | 0.72 |
| **Set to 2.0** | 11.22 | 0.83 | 52.75 | 0.75 |
| **Set to 3.0** | 10.93 | 0.83 | 51.47 | 0.75 |

**In the last of our revised manuscript**

**Line 139 to 141:**

… To avoid unreasonably high Kzmin under stable condition, we set the upper/lower limit value of Kzmin as 2.0/0.01 (covering 93.9% grid values within 0.01-2.0, 2.3% smaller than 0.01). …

**Comment 13:**

l.220: "We need to use a larger Kzmin value to enhance turbulence under a strong stable atmosphere and small or no adjusted Kzmin values under a weak stable or neutral atmosphere." Here there is an explanation of the choice made for the new Kzmin. But there is no physical proof of this choice.

Thank you for your valuable suggestion. We realized that the sentence is not very exact. In the revised manuscript, we further analyze the relation between adjusted Kzmin (fomula 5) and simulated $PM_{2.5}$ bias and try to give a reasonable physical proof. As shown in Figure 5, the new Kzmin scheme enhanced Kzmin values over eastern China, much larger than the default value of 0.01 $m^2 \cdot s^{-1}$. The distribution of the monthly averaged nocturnal Kzmin values exhibited a latitudinal difference with 0.88 $m^2 \cdot s^{-1}$ (0.8-1.3 $m^2 \cdot s^{-1}$) in the NCP and 1.17 $m^2 \cdot s^{-1}$ (1.0-1.5 $m^2 \cdot s^{-1}$) in the YRD, which is within the reasonable Kzmin ranges based on the sensitivity experiments in section 2.3 (Table S1). Also, in Figure R4, the $PM_{2.5}$ bias shows a nonlinear positive correlation with the Kzmin calculated by formula 5 in NCP and YRD. Most of kzmin values (69%) in the NCP are less than 1, while most of Kzmin values (70%) in the YRD are greater than 1. The evidence indicated that the formula 5 is available to reflect the dependency of Kzmin on the landuse and the meteorology with a latitudinal effect. The latitudinal effect could relate to solar radiation, air temperature, cloud, precipitation and landuse in the 2 climate zones (Zhou et al., 2014, Jin et al., 2021) and in mostly extent can be expressed by fomula 5. As such, the formula 5 can reasonable give the dynamic Kzmin values over east China in this study. We have added above sentences into the revise manuscript in line 231 to 236.

**Reference**

Jin, H., Chen, X., Wu, P., Song, C., & Xia, W. (2021). Evaluation of spatial-temporal distribution of precipitation in mainland China by statistic and clustering methods. Atmospheric Research, 262, 105772.

Zhou, L. T., & Huang, R. (2014). Regional differences in surface sensible and latent heat fluxes in China. Theoretical and applied climatology, 116, 625-637.

[Figure]

**Figure 5.** The distribution of Kzmin (unit: $m^2 \cdot s^{-1}$) in EXP_NEW.

[Figure]

**Figure S4.** Scatter plots of the Kzmin and the bias of PM$_{2.5}$.

**In the last of our revised manuscript**

**Line 231 to 236**

Also, in Figure S4, the PM$_{2.5}$ bias shows a nonlinear positive correlation with the Kzmin calculated by fomula 5 in NCP and YRD. Most of kzmin values (69%) in the NCP are less than 1, while most of Kzmin values (70%) in the YRD are greater than 1. The evidence indicated that the formula 5 is available to reflect the dependency of Kzmin on the landuse and the meteorology with a latitudinal effect. The latitudinal effect could relate to solar radiation, air temperature, cloud, precipitation and landuse in the 2 climate zones (Zhou et al., 2014, Jin et al., 2021) and mostly extent can be expressed by fomula 5. As such, the formula 5 can reasonable give the dynamic Kzmin values over east China in this study.

**Comment 14:**

l.223: Please use carefully the term 'climate' (not correct in the present context).

≫Line 223-224: As such, EF can reflect thermal flux features related to climate and the underlying surface in different regions. …

Thank you for this valuable suggestion. The NCP is located in the temperate monsoon climate and is the largest plain area in China with drier soils. The YRD is located in the subtropical monsoon zone, which has a more complex topography and wetter soils (Han et al., 2020). there are distinct features in meteorology/climate and geography, eg. solar radiation, air temperature, cloud, precipitation and landuse, which in mostly extent can be expressed by latent heat flux and sensible heat flux in formula 5 in the two regions (Zhou et al., 2014).

**Reference**

Han, G., Wang, J., Pan, Y., Huang, N., Zhang, Z., Peng, R., ... & Pan, Z. (2020). Temporal and spatial variation of soil moisture and its possible impact on regional air temperature in China. Water, 12(6), 1807.

Zhou, L. T., & Huang, R. (2014). Regional differences in surface sensible and latent heat fluxes in China. Theoretical and applied climatology, 116, 625-637.

**Comment 15:**

l.232: It is stated that: "As most primary pollutants are emitted to the first level in the model", it is not the case of many models. For example, in Europe, anthroogenic emissions are vertically distributed following a vertical profile proposed by EMEP. Fires and dust emissions are often injected following a vertical profile.

≫ Line 232 … As most primary pollutants are emitted to the first level in the model, …

Thank you for pointing out the deficiencies in the description of the manuscript. In our study, the anthropogenic emission inventory that we used is MEIC (in China) and MIX (east of Asia that excluding China). Both MEIC and MIX are divided into five sector emission, including (power, residential, transport, residential and industry emission). According to the vertical profile suggestion by MEIC, residential, transport and residential emission was emitted to the first level in the model. For power emission, it was emitted from second level to seventh level (about 550m). For the industry emission, it was emitted from first level to third level (about 120m). The emission inject height of different sector source was shown in Figure R3a. All residential, transport, residential emission and 50 % industry emission are emitted to the first leverl. Figure R3b is the NO emission in Nanjing(32.0°N,118.4°E). From Figure R3b, we can find that about 83.52% NO was emitted to the first level in the model. There may be something wrong with the presentation of the manuscript. To avoid it, we have revised the text in the revised manuscript line 244.

[Figure]

**Figure R3.** The emission percentage in this study. (a) different setctor emission percentage, (b) NO emission percentage in Nanjing (32.0°N,118.4°E)

**In the last of our revised manuscript in**

**Line 244:**

… As larger proportion of primary pollutants are emitted to the first level in the model, …

**Comment 16:**

In this study PM$_{2.5}$ have different origins. Can a sensitivity experiment diagnose what source (anthropogenic? fires? biogenic? dust? etc.) could be responsible of the observed bias close to the surface?

Thank you for this valuable suggestion. In this study, the input emission inventory are anthropogenic emission inventory (MEIC: in China, index year is 2016; MIX: east of Asia that excluding China, index year 2010) and MEGAN biogenic emission inventory. Fires and dust emissions have not added because there were no significant 2 kind cases occurred in this study period after we checked. We have set two experiment to calculated the contribution of biogenic emission. The result showed that biogenic emission contributed less to the PM$_{2.5}$ concentration. When the anthropogenic emissions inventory is up to date (MEIC, December in 2016) in this study, the incorrect simulation of meteological field could be responsible of the observed bias.

**Table R1.** Mean model performance metrics for and PM$_{2.5}$ (nighttime).

| Case_name | MB | IOA | RMSE | R |
|---|---|---|---|---|
| With Megan | 48.23 | 0.72 | 82.68 | 0.66 |
| Without Megan | 47.89 | 0.72 | 82.17 | 0.66 |

---

## Author Comment (AC2)

Dear reviewer,

Thank you for giving us the opportunity to revise our manuscript (Manuscript Number: egusphere-2023-1089). We appreciate your constructive comments and suggestions, which we have studied carefully while making appropriate revisions on the manuscript. We believe that under your guidance, our manuscript has been substantially improved.

In the following, the reviewer's comments/suggestions are highlighted by gray. The symbol "≫" quotes the original texts in the manuscript. Followed by the comments are our responses in plain text, as well as the respective revisions in the manuscript. Some important revisions are marked with red font.

Thank you again for your constructive comments and suggestions.
Yours sincerely,

Wen Lu, Bin Zhu, and all co-authors.

**Replies to Reviewer**
**General Comments:**
"Parameterized minimum eddy diffusivity in WRF-Chem(v3.9.1.1) for improving $PM_{2.5}$ simulation in the stable boundary layer over eastern China" by Lu et al. proposed a parameterization formula for minimum turbulent diffusivity (Kzmin) and tested the simulations effects for $PM_{2.5}$. The results show that the revised Kzmin parameterization formula improved the $PM_{2.5}$ simulation by improving turbulent diffusion under stable conditions. Weak turbulence in SBL is the key challenge in restricting progress of SBL theory and simulation, the topic of the manuscript is very important. However, the physical logic of the revised Kzmin parameterization formula is questionable, and numerical experiments need to be added. Therefore, I recommend major revision.

We feel great thanks for your professional review work on our article. As you are concerned, there are several problems that need to be addressed. According to your nice suggestions, we have made extensive corrections to our pervious draft, and the detailed corrections are listed below. We think your suggestions and comments have greatly improved our manuscript in science and technical details.

**Specific comments:**
**1)** Line 45, the SBL, weak turbulence and turbulence intermittency are hot topics in studies of atmospheric boundary layer with a lot of papers and progresses, I suggest the citations here keep up with the latest developments.
≫Line 45-46: Studies of the SBL remain insufficient; the SBL is often accompanied by intermittent turbulence and decoupling of the surface and free troposphere (Louis 1979; Grachev et al. 2005).

Thank you for this valuable suggestion. As you say, weak turbulence and turbulence intermittency have gotten a lot of attentions in recent years over the world. We have cited the latest research in the line 44-46 of the revised manuscript.

**In the last of our revised manuscript in Line 44-46:**
… While at present, the mesoscale meteorological numerical models cannot reasonably capture the weak turbulence and turbulence intermittency under SBL (Teixeira et al., 2008, Mahrt et al., 2020, Van der Linden et al., 2020, Jia et al., 2021, Allouche et al., 2022, Ren et al., 2023).

**Reference:**

Allouche, M., Bou-Zeid, E., Ansorge, C., Katul, G. G., Chamecki, M., Acevedo, O., ... & Fuentes, J. D. (2022). The detection, genesis, and modeling of turbulence intermittency in the stable atmospheric surface layer. Journal of the Atmospheric Sciences, 79(4), 1171-1190.

Jia, W., Zhang, X., Zhang, H., & Ren, Y. (2021). Application of turbulent diffusion term of aerosols in mesoscale model. Geophysical Research Letters, 48(11), e2021GL093199.    In page2

Mahrt, L., & Bou-Zeid, E. (2020). Non-stationary boundary layers. Boundary-Layer Meteorology, 177, 189-204.

Ren, Y., Zhang, H., Zhang, L., & Liang, J. Quantitative description and characteristics of submeso motion and turbulence intermittency. Quarterly Journal of the Royal Meteorological Society.

Teixeira, J., Stevens, B., Bretherton, C. S., Cederwall, R., Doyle, J. D., Golaz, J. C., ... & Soares, P. M. (2008). Parameterization of the atmospheric boundary layer: a view from just above the inversion. Bulletin of the American Meteorological Society, 89(4), 453-458.

Van Der Linden, S. J., Van De Wiel, B. J., Petenko, I., Van Heerwaarden, C. C., Baas, P., & Jonker, H. J. (2020). A Businger mechanism for intermittent bursting in the stable boundary layer. Journal of the Atmospheric Sciences, 77(10), 3343-3360.

**2)** Line 49, "Huang et al. (2010)" did not show in Reference list.

≫ … Huang et al. (2010) improved the turbulent fluxes in the SBL by redefining the closure constants and modifying the sensible heat flux prognostic equation. …

We apologize of the incorrect citation. The correct cited year of reference should be 2017. We have revised the text in the line 49-50 of the revised manuscript.

**In the last of our revised manuscript in**

**Line 49-50:**

… Huang et al. (2017) improved the turbulent fluxes in the SBL by redefining the closure constants and modifying the sensible heat flux prognostic equation. …

**Reference**

Huang, Y., & Peng, X. (2017). Improvement of the Mellor–Yamada–Nakanishi–Niino planetary boundary-layer scheme based on observational data in China. Boundary-Layer Meteorology, 162(1), 171-188, 10.1007/s10546-016-0187-0.

**3)** Section 2.1, the basic information of the field experiment was missing. Readers cannot get anything about the field experiment in such simple description now. Which time periods of the first and second data sets used for model validation? All of this information should be added in detailed.

≫Three sets of data were used to evaluate model performance. The first set of data is the hourly ground-based observations of $PM_{2.5}$ mass concentrations in 89 cities obtained from the China National Environmental Monitoring Center and published online (http://106.37.208.233:20035). The second set of data is the 3 h-hourly meteorological factors at 99 ground observation stations in eastern China. The meteorological factors contain 10 m wind speed, 10 m wind direction, and 2 m temperature. The third set of data is the vertical observations of $PM_{2.5}$ and meteorological factors from field experiments by our group in Nanjing. The field experiment was carried out between 27 December 2016 and 31 December 2016 to obtain the 3 h-resolution vertical distribution data of $PM_{2.5}$.

Thank you for this valuable suggestion. We apologize that we miss enough information about our field experiment and data. The time periods of first and second data set that we used is both from December 1, 2016, to December 31, 2016. The third data set is the vertical observations of $PM_{2.5}$ from field experiment. The observation site is located in the northern suburbs of Nanjing. The coordinates and altitude of the observation site are 32.0°N, 118.4°E and approximately 23 m asl, respectively. The $PM_{2.5}$ concentrations were measured by a PDR-1500 fixed on an unmanned aerial vehicle (UAV) platform from December 27, 2016, to December 31, 2016. The details of field experiment please refer to the reference of Shi et al. (2021). 10 profiles (surface to ~1.0km) of $PM_{2.5}$ in SBL were obtained in the field experiment for model evaluation in vertical. We have added the text in the line 75-80 of the revised manuscript.

**In the last of our revised manuscript in**

**Line 75-80:**

The time periods of first and second data set that we used is both from December 1, 2016, to December 31, 2016. The third set of data is the vertical observations of $PM_{2.5}$ from field experiments. The observation site is located in the northern suburbs of Nanjing. The coordinates and altitude of the observation site are 32.0°N, 118.4°E and approximately 23 m asl, respectively. The $PM_{2.5}$ concentrations were measured by a PDR-1500 fixed on an unmanned aerial vehicle (UAV) platform from December 27, 2016, to December 31, 2016. 10 profiles (surface to ~1.0km) of $PM_{2.5}$ in SBL were obtained for model evaluation in vertical.

**4)** Section 2.3, how many haze cases did the numerical experiments choose? I did not see any introduction about the time periods or haze cases through the manuscript. Or only one case form 27 December 2016 to 31 December 2016? Can the reliability of the results be confirmed by more haze cases? The introductions on sensitivity experiments were also missing.

Thank you for this valuable suggestion. The observation of 8 cities was used to show the haze events in this study. There locations are Baoding (38.87°N, 115.48°E); Dezhou (37.45°N, 116.32°E), Jinan (36.61°N, 116.99°E), Zaozhuang (35.10°N, 117.45°E), Suqian (33.95°N, 118.29°E), Nanjing (32.0°N, 118.4°E); Liyang (31.4°N, 119.46°E), and Hangzhou (30.29°N,120.16°E). Daily mean concentration statistics for each station was shown in Table R1. Three wider regional pollution events (the concentration of $PM_{2.5}$ exceeds 115 ug·m$^{-3}$ and lasts for more than 2 days) was occurred in December 2016. The time range of each pollution events are haze (December 1 to December 9), haze2 (December 16 to December 23), and haze3 (December 28 to December 31). We added the observed spatial and temporal variations of $PM_{2.5}$ in Figure S3 of the revised supplement and line 97-100 of the revised manuscript. As for the reliability of the results about the haze cases, the time series of simulated and observed $PM_{2.5}$ was compared in Figure S2 and the discussion is given in Comment 10, and the new Kzmin has better performance in capturing the temporal evolution of the three haze events mentioned in Figure S3.

[Figure]

**Figure S3.** The observed spatial and temporal variations of $PM_{2.5}$ mass concentration (μg·m$^{-3}$) along the latitudes of the 8 observation sites from north to south. The grey boxes represent three haze events (concentration exceeds 115 ug·m$^{-3}$ and lasts for more than 48 hours). The white color shaded is the missing data.

**Table R1.** Daily mean concentration statistics for each station

| Standard | Baoding | Dezhou | Jinan | Zaozhuang | Suqian | Nanjing | Liyang | Hangzhou |
|---|---|---|---|---|---|---|---|---|
| >35 ug·m$^{-3}$ | 30 | 30 | 31 | 30 | 28 | 23 | 26 | 26 |
| >75 ug·m$^{-3}$ | 27 | 28 | 22 | 22 | 18 | 17 | 18 | 19 |

As for the sensitivity experiments, the default Kzmin value in YSU scheme is 0.01 m$^2$·s$^{-1}$. We set the fixed Kzmin value as 0.3, 0.5, 0.8, 1.0, 1.3, 1.5, 1.8, 2.0 in sensitivity experiments. The other setting is same as the description in Section 2.2. Then, we divided the sites in the simulation area into northern and southern sites for statistics, and the statistical results are shown in Table S1. The introductions on sensitivity experiments were added in the line 121-125 of the revised manuscript.

**In the last of our revised manuscript**
**Line 97-100:**
… The simulation started at 00:00 UTC on 28 November 2016 and ended at 00:00 UTC on 1 January 2017. To eliminate the effect of initial conditions, we set the first 3 d as the spin-up period (Napelenok et al., 2008). Three wider regional pollution events (the concentration of PM$_{2.5}$ exceeds 115 ug·m$^{-3}$ and lasts for more than 2 days) was occurred over eastern Chian in December 2016 (Figure S3).

**Line 121-125:**
… Therefore, we set the fixed Kzmin value as 0.3, 0.5, 0.8, 1.0, 1.3, 1.5, 1.8, 2.0 in sensitivity experiments (shown in Table S1, the available values are marked in red). We found that the reasonable Kzmin values for winter aerosol simulations have latitudinal difference in eastern China (the north of eastern China, NCP: 0.8 to 1.3 m$^2$·s$^{-1}$, the Yangtze River Delta, YRD:1.0 to 1.5 m$^2$·s$^{-1}$). …

**5)** Line 105, some studies had revealed that the turbulent characteristics of PM$_{2.5}$ are different with heat, this information should be clarified.
≫Line 103-104: ... The turbulent mixing process of pollutants is considered to be similar to that of heat, which supposes the turbulent diffusion of particles and heat is identical.

Thank you for this valuable suggestion. It should be noted that here just states how the turbulent diffusion coefficients of PM$_{2.5}$ in the WRF-Chem model are obtained. In recent, there have been observational study (Ren et al., 2021) showed that the relationship between turbulent difussion coefficient of PM$_{2.5}$ (Kc) and turbulent difussion coefficient of heat (K$_H$) cannot be completely determined. In this study, we still treat particles as a scalar. To avoid ambiguity, we have revised the text in line 107-108 of revised manuscript.

**In the last of our revised manuscript in**
**Line 107-108:**
…The value of the diffusion coefficient of chemical compositionis is assumed to be equal to the value of the heat diffusion coefficient in Chem module (Jia et al, 2021b). …

**6)** Line 111, () was missing in "Noh et al., 2003".
≫Line 111: ... According to the Prandtl number as in Noh et al., 2003:

Thank you for this valuable suggestion. We have revised the text in revised manuscript line 115.

**In the last of our revised manuscript in**

**Line 115:**

… According to the Prandtl number as in Noh et al. (2003):

**7)** The parameterization of the new value of Kzmin was proposed abruptly without sufficient physical discussion. Line 129, the authors described "We assume that the value of EF can be used to characterize Kzmin in different regions." Why did you propose this assumption? In other words, what is the physical meaning behind the formula 5? Why is it set up in this form? Line 223, "After setting the adjustment factor value to 1.0 in formula 3", why did you set an adjustment factor if you "assume that the value of EF can be used to characterize Kzmin in different regions"? and What is the basis for setting 1.0 as the adjustment factor? Very confusing, was it formula 5 or 3 in Line 223? Under stable conditions, because of the high value of EF, the values of Kzmin were at least 100 times larger than under unstable conditions, Kh might be close to 2, was it reasonable? I highly doubt the physical rationality. Based on your formula 5, Kh under stable conditions might larger than under unstable conditions. Anyway, the formula 5 you proposed was the core of this manuscript, more physical explanations are needed.

≫Line 129: We assume that the value of EF can be used to characterize Kzmin in different regions.

≫Line 223-225: ... As such, EF can reflect thermal flux features related to climate and the underlying surface in different regions. After setting the adjustment factor value to 1.0 in formula 3 …

Thank you for your valuable suggestion. In the revised manuscript, we further analyze the relation between adjusted Kzmin (formula 5) and simulated $PM_{2.5}$ bias and try to give a reasonable physical proof.

Firstly, as shown in Figure 5, the new Kzmin scheme enhanced Kzmin values over eastern China, much larger than the default value of 0.01 $m^2 \cdot s^{-1}$. The distribution of the monthly averaged nocturnal Kzmin values exhibited a latitudinal difference with 0.88 $m^2 \cdot s^{-1}$ (0.8-1.3 $m^2 \cdot s^{-1}$) in the NCP and 1.17 $m^2 \cdot s^{-1}$ (1.0-1.5 $m^2 \cdot s^{-1}$) in the YRD, which is within the reasonable Kzmin ranges based on the sensitivity experiments in section 2.3 (Table S1). Also, in figure S6, the $PM_{2.5}$ bias shows a nonlinear positive correlation with the Kzmin in NCP and YRD. Most of kzmin values (69%) in the NCP are less than 1, while most of Kzmin values (70%) in the YRD are greater than 1. The evidence indicated that the formula 5 is available to reflect the dependency of Kzmin on the landuse and the meteorology with a latitudinal effect. The latitudinal effect could relate to solar radiation, air temperature, cloud, precipitation and landuse in the 2 climate zones (Zhou et al., 2014, Jin et al, 2021) and in mostly extent can be expressed by 5. As such, the formula 5 can reasonable give the dynamic Kzmin values over east China in this study. We have added above sentences into the revise manuscript in line 231 to 236.

[Figure]

**Figure 5.** The distribution of Kzmin (unit: $m^2 \cdot s^{-1}$) in EXP_NEW.

[Figure]

**Figure S4.** Scatter plots of the Kzmin and the bias of $PM_{2.5}$.

Secondly, The turbulent diffusion coefficient $K_h$ was calculated following fomula 3: $K_h = K_m/P_r + Kzmin$,

where $K_m$ is momentum mixing coefficient, Pr is the prandtl number.

The default value of the Kzmin in YSU PBL scheme is 0.01 $m^2 \cdot s^{-1}$. The aim of our work is to improve the simulation for the stable condition. Therefore, we only parameterized the Kzmin value during stable condition. While for the unstable, we let Kzmin consistent with the default settings (0.01 $m^2 \cdot s^{-1}$). we find that the new adjusted Kh were 1.35 $m^2 \cdot s^{-1}$ in NCP and 2.03 $m^2 \cdot s^{-1}$ in YRD, which are much lower than the $K_h$ in daytime. For example, the values of $K_h$ in EXP_BASE and EXP_NEW in Nanjing (32.0°N,118.4°E) was shown in Figure R1. There is no $K_h$ value at stable condition larger than that in unstable condition. The simulation results of Du et al., (2020) and Jia et al., (2021) indicated that $K_h$ great than 2.0 is a general value under SBL in eastern China.

[Figure]

**Figure R1.** The turbulent diffusion coefficient in EXP_NEW and EPX_BASE(unit: $m^2 \cdot s^{-1}$) in Nanjing. The yellow shaded represent the stable condition (Richardson number>0).

**Reference**

Du, Q., Zhao, C., Zhang, M., Dong, X., Chen, Y., Liu, Z., ... & Miao, S. (2020). Modeling diurnal variation of surface $PM_{2.5}$ concentrations over East China with WRF-Chem: Impacts from boundary-layer mixing and anthropogenic emission. Atmospheric Chemistry and Physics, 20(5), 2839-2863, 10.5194/acp-20-2839-2020.

Jia, W., & Zhang, X. (2021). Impact of modified turbulent diffusion of $PM_{2.5}$ aerosol in WRF-Chem simulations in eastern China. Atmospheric Chemistry and Physics, 21(22), 16827-16841, 10.5194/acp-21-16827-2021.

Jin, H., Chen, X., Wu, P., Song, C., & Xia, W. (2021). Evaluation of spatial-temporal distribution of precipitation in mainland China by statistic and clustering methods. Atmospheric Research, 262, 105772.

Zhou, L. T., & Huang, R. (2014). Regional differences in surface sensible and latent heat fluxes in China. Theoretical and applied climatology, 116, 625-637.

**In the last of our revised manuscript**

**Line 231 to 236**

… Also, in Figure S4, the PM$_{2.5}$ bias shows a nonlinear positive correlation with the Kzmin calculated by fomular 5 in NCP and YRD. Most of kzmin values (69%) in the NCP are less than 1, while most of Kzmin values (70%) in the YRD are greater than 1. The evidence indicated that the formula 5 is available to reflect the dependency of Kzmin on the landuse and the meteorology with a latitudinal effect. The latitudinal effect could relate to solar radiation, air temperature, cloud, precipitation and landuse in the 2 climate zones (Zhou et al., 2014, Jin et al., 2021) and in mostly extent can be expressed by fomular 5. As such, the formula 5 can reasonable give the dynamic Kzmin values over east China in this study. …

**8)** Line 134, "the value calculated by formula 1", was it formula 5?

≫Line 133-134: ... the Kzmin value was set to the value calculated by formula 1. …

We apologize for this mistake. We have revised the text in the line 138 of the revised manuscript.

**In the last of our revised manuscript in**

**Line 138:**

⋯ the Kzmin value was set to the value calculated by formula 5 …

**9)** Line 140, the formulas of MB, IOA, RMSE, R, NMB and NME should be clarified at somewhere appropriate.

≫Line 140-141: ⋯ model performance metrics (MB, mean bias; IOA, index of agreement; RMSE, root mean square error; R: correlation coefficient, NMB: normalized mean bias, NME: normalized mean error) were used …

The definition of the metrics that we used in our manuscript are added according to (revised supplement line 10-17)

**In the last of our revised supplement**

**Line 10-17:**

Mean Bias: $MB = \frac{1}{N}\sum_{i=1}^{N}(M_i - O_i)$

Index Of Agreement: $IOA = 1 - \frac{\sum_{i=1}^{N}(M_i - O_i)^2}{\sum_{i=1}^{N}(|M_i - \bar{M}| + |O_i - \bar{O}|)^2}$

Root Mean Square Error: $RMSE = \sqrt{\frac{1}{N}\sum_{i=1}^{N}(M_i - O_i)^2}$

Correlation Coefficient: $R = \frac{Cov(x,y)}{\sqrt{x}\sqrt{y}}$

Normalized Mean Bias: $NMB = \frac{\sum_{i=1}^{N}(M_i - O_i)}{\sum_{i=1}^{N}O_i} \times 100\%$

Normalized Mean Error: $NME = \frac{\sum_{i=1}^{N}|M_i - O_i|}{\sum_{i=1}^{N}O_i} \times 100\%$

**10)** The mean model performance in Table 1 and Table S1 means the mean performance from several cases or one case in whole domain? Another key issue is that I did not see any comparison of the simulated and observed PM$_{2.5}$ time series.

Thank you for this valuable suggestion. The mean model performace in Table1 and Table S1 is the mean performance in the whole month (December 2016). We added the time series comparision of observed and similated PM$_{2.5}$ into the supplement. The simulation of four citys (2 north city; 2 south city) was used to demonstrate the improvements. There locations are

Zhengzhou (34.75°N, 113.64°E); Jining (35.43°N, 116.63°E) in north of China and Hefei (31.94°N, 117.27°E); Nanjing (32.2°N, 118.8°E) in YRD. Three to four pollution events (the concentration of PM$_{2.5}$ exceeds 115 ug·m$^{-3}$ and lasts for more than 2 days, shadw in yellow) occurred in each city in December 2016. The overestimation of EXP_BASE is obvious, especially during stable condition, e.g., at night, and during heavy pollution events. The model result in EXP_NEW is improved by using our parameterized Kzmin and close to the observations. The metrics such as MB and IOA has significant improvement. It's worth noting that both two schemes underestimate extreme high peak concentrations of PM$_{2.5}$ (such as Zhengzhou on 19 December, the concentration greater than 600 ug·m$^{-3}$), which may be due to the poor ability to extremely heave pollution simulation in existing mesoscale models. The hourly time-series was added in the Figure S2 of the revised supplement.

**In the last of our revised supplement**

[Figure]

**Figure S2.** Time series of PM$_{2.5}$ concentration in Zhengzhou, Jining, Hefei and Nanjing in December 2016. The grey dots, red lines and blue lines represent the results of observation, EXP_BASE and EXP_NEW, respectively. The yellow shaded represent haze events (the concentration of PM$_{2.5}$ exceeds 115 ug·m$^{-3}$ and lasts for more than 48 hours) in each city. The metrics (MB, IOA and R) was calculated by the full day (daytime and nighttime) data.

**11)** Line 238, "Figure 5d, 5e" means "Figure 6d, 6e"? same mistake in line 239, Figure 5 f.

≫Line 238: ... between EXP_BASE and EXP_NEW (Figure 5d, 5e). ...

≫Line 239: ... Figure 5f shows that the difference ...

We apologize for the mistakes. We have revised the text accordingly in the revised manuscript line 250, line 251.

**In the last of our revised manuscript in**

**Line 250:**

… between EXP_BASE and EXP_NEW (Figure 6d, 6e). …

**Line 251:**

… Figure 6f shows that the difference …

**12)** Line 265, "Figure 7. The distribution of difference …", you mean the $PM_{2.5}$ concentrations difference?
≫Line 265: Figure 7. The distribution of difference in difference process contribution.

We apologize for the unclear description in the figure caption for Figure 7. It is the distribution of process contribution differences of VMIX, ADV, AERO, and NET in model simulations. In section 3.3, we use process analysis to determine the key process of the improvement. For example, the vertical mixing (VMIX) value in EXP_BASE is represent the contribution of turbulent diffusion process to the change of $PM_{2.5}$. In Figure 6a, it is obvious that VMIX contribution on the surface is negetive, which means turbulance difussion process reduces the surface $PM_{2.5}$ concentrations. The VMIX value of EXP_BASE minus the VMIX value of EXP_NEW is negative on the surface and positive on the upper BL. It indicates that EXP_NEW enhanced surface turbulance diffusion process compared to the EXP_BASE and diffused more $PM_{2.5}$ into the high altitude and thus increasing the concentration of $PM_{2.5}$ in the high altitude. We have revised the description between line 276-277 in the revised manuscript.

**In the last of our revised manuscript in**
**Line 276-277:**
**Figure 7.** The distribution of PM$_{2.5}$ process contribution difference between EXP_NEW and EXP_BASE (unit: ug·m$^{-3}$·h$^{-1}$). a: VMIX, b: ADV, c: AERO d: NET=VMIX+ADV+AERO.

**13)** Writing needs to be further improved.

Thank you for this valuable suggestion. We made revisions to the paper and received help from the English rewriting Agency AJE. We hope that our revised paper will be approved by you!